# OPTIMIZING $(L_0, L_1)$-SMOOTH FUNCTIONS BY GRADIENT METHODS

**Daniil Vankov**[*]
Arizona State University
dvankov@asu.edu

**Anton Rodomanov**
CISPA[†]
anton.rodomanov@cispa.de

**Angelia Nedić**
Arizona State University
angelia.nedich@asu.edu

**Lalitha Sankar**
Arizona State University
lsankar@asu.edu

**Sebastian U. Stich**
CISPA[†]
stich@cispa.de

## ABSTRACT

We study gradient methods for optimizing $(L_0, L_1)$-smooth functions, a class that generalizes Lipschitz-smooth functions and has gained attention for its relevance in machine learning. We provide new insights into the structure of this function class and develop a principled framework for analyzing optimization methods in this setting. While our convergence rate estimates recover existing results for minimizing the gradient norm in nonconvex problems, our approach significantly improves the best-known complexity bounds for convex objectives. Moreover, we show that the gradient method with Polyak stepsizes and the normalized gradient method achieve nearly the same complexity guarantees as methods that rely on explicit knowledge of $(L_0, L_1)$. Finally, we demonstrate that a carefully designed accelerated gradient method can be applied to $(L_0, L_1)$-smooth functions, further improving all previous results.

## 1 INTRODUCTION

In this paper, we focus on the deterministic unconstrained optimization problem

$$f^* := \min_{x \in \mathbb{R}^d} f(x), \tag{1}$$

where $f: \mathbb{R}^d \to \mathbb{R}$ is an $(L_0, L_1)$-smooth function. With the rise of deep learning, ensuring efficient convergence has become increasingly critical. Traditional optimization methods, such as the gradient descent method and its variants, often rely on assumptions like Lipschitz-smoothness to guarantee convergence rates. However, in modern machine learning problems, these assumptions might be too restrictive, especially when optimizing deep neural network models.

Experiments in (Zhang et al., 2019) demonstrated that the Hessian norm correlates with the gradient norm of the loss when training neural networks. This observation led the authors to propose $(L_0, L_1)$-smoothness, a more realistic smoothness assumption that generalizes classical Lipschitz smoothness. They also analyzed the gradient method (GM) with fixed, normalized, and clipped stepsizes for nonconvex optimization, showing that normalized and clipped methods perform more favorably in the new setting. In recent years, numerous studies have investigated optimization methods under $(L_0, L_1)$-smoothness. However, despite growing interest, existing convergence results remain suboptimal in key cases, and the theoretical analysis of these methods is still incomplete.

To address this gap, this work explores the properties of $(L_0, L_1)$-smooth functions and investigates gradient methods for their optimization.

**Contributions.** Our main contributions can be summarized as follows:

---

[*]Part of the work was done while DV visited the CISPA Helmholtz Center for Information Security.
[†]CISPA Helmholtz Center for Information Security.

- In Section 2, we provide novel results and insights into the $(L_0, L_1)$-smooth class by (i) presenting new examples and operations preserving $(L_0, L_1)$-smoothness, and (ii) deriving new properties of this function class, leading to tighter bounds on the function value and its gradient. In Section 3, we propose new, intuitive step sizes that directly follow from minimizing our tighter upper bound on the function growth. We also discuss the relation between these stepsizes and those used in the normalized and clipped gradient methods.

- For nonconvex functions, our gradient methods achieve the best-known $\mathcal{O}(\frac{L_0 F_0}{\epsilon^2} + \frac{L_1 F_0}{\epsilon})$ complexity bound for finding an $\epsilon$-stationary point, where $F_0 := f(x_0) - f^*$ is the function residual at the initial point (Theorem 3.1). For convex problems, we significantly improve existing results by showing that an $\epsilon$-approximate solution in terms of the function value can be found in at most $\mathcal{O}(\frac{L_0 R^2}{\epsilon} + L_1 R \ln \frac{F_0}{\epsilon})$ gradient queries, where $R = \|x_0 - x^*\|$ is the initial distance to a solution (Theorem 3.2).

- We also study two other methods: normalized gradient method (NGM) and gradient method with Polyak stepsizes (PS-GM), neither of which requires the knowledge of $(L_0, L_1)$. For both methods, we show that they enjoy the $\mathcal{O}\big(\frac{L_0 R^2}{\epsilon} + [L_1 R]^2\big)$ complexity (see Theorems 4.1 and 5.1).

- Finally, in Section 6, we prove the $\nu \mathcal{O}\big(\sqrt{\frac{L_0 R^2}{\epsilon}} + \lceil (L_1 R)^{2/3}\rceil \lceil \ln \frac{F_0}{\epsilon}\rceil\big)$ complexity bound for the Accelerated Gradient Method with Small-Dimensional Relaxation (AGMsDR), where $\nu \geq 1$ denotes the number of oracle queries required for one-dimensional minimization of the objective over an interval (see Theorem 6.2).

In contrast to other results in the literature, all our complexity bounds neither depend on the initial gradient norm nor have an exponential dependency on $L_0$ or $L_1$.

**Related work.**    Following the introduction of the $(L_0, L_1)$-class by Zhang et al. (2019), subsequent works have explored other smoothness generalizations and analyzed gradient methods under these new assumptions. Chen et al. (2023) introduced the $\alpha$-asymmetric class, relaxing the assumption on twice differentiability and allowing a sublinear growth on the norm of a gradient. In (Li et al., 2023), authors went further and proposed the weakest $(r, l)$-smooth class, which allows even quadratic growth of the norm of the Hessian with respect to the norm of the gradient. Despite the generality of this assumption, there are still some issues and open questions regarding the existing results even for the basic $(L_0, L_1)$-smooth class.

In (Zhang et al., 2020), the authors analyzed the clipped GM with momentum and improved the complexity bound with respect to $(L_0, L_1)$. Using the right choice of clipping parameters, Koloskova et al. (2023) proved, for nonconvex and convex problems respectively, the $\mathcal{O}(\frac{L_0 F_0}{\epsilon^2} + \frac{L_1 F_0}{\epsilon})$ and $\mathcal{O}(\frac{L_0 R}{\epsilon} + \sqrt{\frac{L}{\epsilon}} L_1 R^2)$ complexity bounds, where $L$ is the standard Lipschitz-smoothness constant. For convex problems, Li et al. (2023) proposed an (asymptotically) faster accelerated gradient method whose complexity is $\mathcal{O}\big((L_1^2 R^2 + \frac{L_1^2 F_0}{L_0} + 1)\sqrt{\frac{F_0 + L_0 R^2}{\epsilon}}\big)$[1]. Several works have studied adaptive optimization methods that do not require the $(L_0, L_1)$ parameters to be known. Faw et al. (2023); Wang et al. (2023) studied convergence rates for AdaGrad for stochastic nonconvex problems. Hübler et al. (2024) proposed a gradient method with the backtracking line search and showed the $\mathcal{O}(\frac{L_0 F_0}{\epsilon^2} + \frac{L_1^2 F_0^2}{\epsilon^2})$ complexity bound for nonconvex problems. For convex problems, Takezawa et al. (2024) proved that the PS-GM method enjoys the complexity of $\mathcal{O}(\frac{L_0 R}{\epsilon} + \sqrt{\frac{L}{\epsilon}} L_1 R^2)$.

A closely related paper that appeared online independently during the finalization of our manuscript is (Gorbunov et al., 2024). The authors introduce a new stepsize selection strategy for gradient methods on convex $(L_0, L_1)$-smooth functions, called "smooth clipping," which, up to absolute constants, coincides with one of our formulas. Their proof techniques differ from ours, resulting in a slightly worse complexity bound of $\mathcal{O}(\frac{L_0 R^2}{\epsilon} + [L_1 R]^2)$ compared to our $\mathcal{O}(\frac{L_0 R^2}{\epsilon} + L_1 R \ln \frac{F_0}{\epsilon})$, particularly when the initial function value is reasonably bounded (see Section 3). They also show that PS-GM achieves the same efficiency bound as in our work. Additionally, the authors present an accelerated method with complexity $\mathcal{O}(1) \exp(\mathcal{O}(1) L_1 R) \sqrt{\frac{L_0 R^2}{\epsilon}}$, and extend their analysis to strongly convex, stochastic and adaptive methods. In contrast, our work has a slightly different focus, offering

---

[1]See Section F.

deeper insights by deriving principled stepsize formulas, analyzing nonconvex functions, studying nornalized gradient methods, and developing a superior acceleration scheme with significantly better complexity. Moreover, our proof techniques differ from those in (Gorbunov et al., 2024).

## 2 DEFINITION AND PROPERTIES OF $(L_0, L_1)$-SMOOTH FUNCTIONS

In this section, we state our assumptions and discuss important properties of generalized smooth functions. We start with defining our main assumption on $(L_0, L_1)$-smooth functions.

Throughout this paper, unless specified otherwise, we use the standard inner product $\langle \cdot, \cdot \rangle$ and the standard Euclidean norm $\| \cdot \|$ for vectors, and the standard spectral norm $\| \cdot \|$ for matrices. We also assume that problem (1) admits a solution.

**Definition 2.1.** A twice continuously differentiable function $f \colon \mathbb{R}^d \to \mathbb{R}$ is called $(L_0, L_1)$-*smooth* (for some $L_0, L_1 \geq 0$) if it holds that

$$\|\nabla^2 f(x)\| \leq L_0 + L_1 \|\nabla f(x)\|, \qquad \forall x \in \mathbb{R}^d. \tag{2}$$

The class of $(L_0, L_1)$-smooth functions is a wide family which includes the class of Lipschitz-smooth functions, and was introduced in (Zhang et al., 2019). For twice differentiable functions, this definition is equivalent to that of $\alpha$-symmetric functions with $\alpha = 1$ proposed in (Chen et al., 2023). Since any $\alpha$-symmetric twice differentiable function is also $(L_0, L_1)$-smooth with a different choice of parameters, all our subsequent results hold for $\alpha$-symmetric functions as well.

For the purpose of analysis of the methods, we provide an alternative and more useful first-order characterization of the class of $(L_0, L_1)$-smooth functions.

**Lemma 2.2.** *Let $f$ be a twice continuously differentiable function, Then, $f$ is $(L_0, L_1)$-smooth if and only if any of the following inequalities holds for any $x, y \in \mathbb{R}^d$:*[2]

$$\|\nabla f(y) - \nabla f(x)\| \leq (L_0 + L_1 \|\nabla f(x)\|) \frac{e^{L_1 \|y - x\|} - 1}{L_1}, \tag{3}$$

$$|f(y) - f(x) - \langle \nabla f(x), y - x \rangle| \leq (L_0 + L_1 \|\nabla f(x)\|) \frac{\phi(L_1 \|y - x\|)}{L_1^2}, \tag{4}$$

*where $\phi(t) \coloneqq e^t - t - 1$ ($t \geq 0$).*

The proof of Lemma 2.2 can be found in Section A.1. It is worth noting that inequality (3) is stronger than that from (Zhang et al., 2020, Corollary A.4). The bound in inequality (4) is tighter than those presented in previous works (see, for example, Lemma A.3 in (Zhang et al., 2020), Lemma 8 in (Hübler et al., 2024)). These tighter estimates allow us to construct gradient methods in the sequel.

In our analysis, we often use certain properties of the function $\phi$ and its conjugate[3] $\phi_*$, which we summarize in the following lemma (see Section A.7 for the proof).

**Lemma 2.3.** *The following statements for the function $\phi(t) = e^t - t - 1$ hold true:*

1. $\phi(t) \leq \frac{t^2}{2(1 - \frac{t}{3})}$ *for all $t \in [0, 3)$ and $\phi(t) \leq \frac{t^2}{2} e^t$ for all $t \geq 0$.*

2. $\phi_*(\gamma) \coloneqq \max_{t \geq 0} \{\gamma t - \phi(t)\} = (1 + \gamma) \ln(1 + \gamma) - \gamma$ *for any $\gamma \geq 0$.*

3. $\frac{\gamma^2}{2 + \gamma} \leq \phi_*(\gamma) \leq \frac{\gamma^2}{2}$ *for all $\gamma \geq 0$.*

When $f$ is also convex, we have the following useful inequalities (see Section A.2 for the proof).

---

[2]Hereinafter, for $L_1 = 0$ and any $t \geq 0$, we assume that $\frac{e^{L_1 t} - 1}{L_1} \equiv t$, $\frac{\phi(L_1 t)}{L_1^2} \equiv \frac{1}{2} t^2$, etc., which are the limits of these expressions when $L_1 \to 0$; $L_1 > 0$.

[3]The conjugate function is defined in the standard way: $\phi_*(\gamma) \coloneqq \max_{t \geq 0} \{\gamma t - \phi(t)\}$.

**Lemma 2.4.** *Let $f$ be a convex $(L_0, L_1)$-smooth nonlinear[4] function. Then, for any $x, y \in \mathbb{R}^d$,*

$$f(y) \geq f(x) + \langle \nabla f(x), y - x \rangle + \frac{L_0 + L_1 \|\nabla f(y)\|}{L_1^2} \phi_* \left( \frac{L_1 \|\nabla f(y) - \nabla f(x)\|}{L_0 + L_1 \|\nabla f(y)\|} \right), \quad (5)$$

$$\langle \nabla f(x) - \nabla f(y), x - y \rangle \geq \frac{L_0 + L_1 \|\nabla f(y)\|}{L_1^2} \phi_* \left( \frac{L_1 \|\nabla f(y) - \nabla f(x)\|}{L_0 + L_1 \|\nabla f(y)\|} \right)$$

$$+ \frac{L_0 + L_1 \|\nabla f(x)\|}{L_1^2} \phi_* \left( \frac{L_1 \|\nabla f(y) - \nabla f(x)\|}{L_0 + L_1 \|\nabla f(x)\|} \right), \quad (6)$$

*where $\phi_*$ is the function from Lemma 2.3.*

Lemma 2.4 is a generalization of (Nesterov, 2018, Theorem 2.1.5) to $(L_0, L_1)$-smooth functions, and matches it when $L_1 = 0$ (since $\frac{1}{L_1^2} \phi_*(L_1 \alpha) \to \frac{1}{2} \alpha^2$ as $L_1 \to 0$). Moreover, using Lemma 2.3, we can simplify the lower bound in (5).

**Corollary 2.5.** *Let $f$ be a convex $(L_0, L_1)$-smooth nonlinear function. Then, for any $x, y \in \mathbb{R}^d$,*

$$f(y) \geq f(x) + \langle \nabla f(x), y - x \rangle + \frac{\|\nabla f(y) - \nabla f(x)\|^2}{2(L_0 + L_1 \|\nabla f(y)\|) + L_1 \|\nabla f(y) - \nabla f(x)\|}. \quad (7)$$

## 3 GRADIENT METHOD

Having established a few important properties of an $(L_0, L_1)$-smooth function $f$, we now turn our attention to the *gradient method* (GM) for minimizing such a function:

$$x_{k+1} = x_k - \eta_k \nabla f(x_k), \qquad k \geq 0, \quad (8)$$

where $x_0 \in \mathbb{R}^d$ is a starting point and $\eta_k \geq 0$ are certain stepsizes.

We start with showing that the gradient update rule (8) and the "right" formula for the stepsize $\eta_k$ both naturally arise from the classical idea in optimization theory—choosing the next iterate $x_{k+1}$ by minimizing the global upper bound on the objective constructed around the current iterate $x_k$ (see (Nesterov, 2018)). Indeed, let $x \in \mathbb{R}^d$ be the current point, and let $a := L_0 + L_1 \|\nabla f(x)\| > 0$. According to (4), for any $y \in \mathbb{R}^d$,

$$f(y) \leq f(x) + \langle \nabla f(x), y - x \rangle + \frac{a}{L_1^2} \phi(L_1 \|y - x\|).$$

Our goal is to minimize the right-hand of the above inequality in $y$. Since the last term in this bound depends only on the norm of $y - x$, the optimal point $y^* = T(x)$ is the result of the gradient step $T(x) = x - r^* \frac{\nabla f(x)}{\|\nabla f(x)\|}$ for some $r^* \geq 0$ ensuring the following progress in decreasing the function value:

$$f(x) - f(T(x)) \geq \max_{r \geq 0} \left\{ \|\nabla f(x)\| r - \frac{a}{L_1^2} \phi(L_1 r) \right\} = \frac{a}{L_1^2} \phi_* \left( \frac{L_1 \|\nabla f(x)\|}{a} \right),$$

where $\phi_*$ is the conjugate function to $\phi$ (see Lemma 2.3). Furthermore, $r^*$ is exactly the solution of the above optimization problem, satisfying $L_1 \|\nabla f(x)\| = a \phi'(L_1 r^*)$. Solving this equation, using $(\phi')^{-1}(\gamma) = \phi_*'(\gamma) = \ln(1 + \gamma)$, we obtain $r^* = \frac{1}{L_1} \phi_*' \left( \frac{L_1 \|\nabla f(x)\|}{a} \right) = \frac{1}{L_1} \ln(1 + \frac{L_1 \|\nabla f(x)\|}{a})$.

The above considerations lead us to the following *optimal* choice of stepsizes in (8):

$$\boxed{\eta_k^* = \frac{1}{L_1 \|\nabla f(x_k)\|} \ln \left( 1 + \frac{L_1 \|\nabla f(x_k)\|}{L_0 + L_1 \|\nabla f(x_k)\|} \right),} \qquad k \geq 0, \quad (9)$$

resulting in the following progress in decreasing the objective:

$$f(x_k) - f(x_{k+1}) \geq \frac{L_0 + L_1 \|\nabla f(x_k)\|}{L_1^2} \phi_* \left( \frac{L_1 \|\nabla f(x_k)\|}{L_0 + L_1 \|\nabla f(x_k)\|} \right) := \Delta_k. \quad (10)$$

---

[4]According to Lemma 2.2, this means that $L_0 + L_1 \|\nabla f(x)\| > 0$ for any $x \in \mathbb{R}^d$.

The above expression for $\Delta_k$ is quite cumbersome but, in fact, it behaves as the simple fraction $\frac{\|\nabla f(x_k)\|^2}{L_0 + L_1 \|\nabla f(x_k)\|}$. More precisely, from Lemma 2.3(3), we see that

$$\frac{\|\nabla f(x_k)\|^2}{2L_0 + 3L_1 \|\nabla f(x_k)\|} \leq \Delta_k \leq \frac{\|\nabla f(x_k)\|^2}{2(L_0 + L_1 \|\nabla f(x_k)\|)}.$$

Thus, there is not much point in keeping the complicated expression (10) and we can safely simplify it as follows:

$$f(x_k) - f(x_{k+1}) \geq \frac{\|\nabla f(x_k)\|^2}{2L_0 + 3L_1 \|\nabla f(x_k)\|}. \tag{11}$$

Interestingly, we can also arrive at exactly the same bound (11) by using a simpler choice of step-sizes. Specifically, replacing $\ln(1 + \gamma)$ with its lower bound $\frac{2\gamma}{2+\gamma}$ (which is responsible for the inequality in Lemma 2.3(3) that we used to simplify (10) into (11)), we obtain the following *simplified stepsizes*:

$$\boxed{\eta_k^{\mathrm{si}} = \frac{1}{L_0 + \frac{3}{2}L_1 \|\nabla f(x_k)\|},} \qquad k \geq 0. \tag{12}$$

With this choice, the iterates of method (8) still satisfy (11) (see Lemma B.1).

Further, note that, up to absolute constants, stepsize (12) acts as $\frac{1}{\max\{L_0, L_1\|\nabla f(x_k)\|\}} = \min\{\frac{1}{L_0}, \frac{1}{L_1\|\nabla f(x_k)\|}\}$, which is the so-called clipping stepsize used in many previous works (Zhang et al., 2019; 2020; Koloskova et al., 2023). Thus, with the right choice of absolute constants, we can expect the corresponding clipping stepsizes, to satisfy a similar inequality to (11). This is indeed the case, and we can show, in particular, that the *clipping stepsizes*

$$\boxed{\eta_k^{\mathrm{cl}} = \min\left\{\frac{1}{2L_0}, \frac{1}{3L_1\|\nabla f(x_k)\|}\right\},} \qquad k \geq 0, \tag{13}$$

do satisfy (11) although with slightly worse absolute constants (see Lemma B.1).

We have thus demonstrated in this section that clipping stepsizes (13) are simply a convenient approximation of the optimal stepsizes (9), ensuring a similar bound on the objective progress. This observation seems to be a new insight into clipping stepsizes which has not been previously explored in the literature.

It is not difficult to see that the three stepsizes we introduced in this section satisfy

$$\eta_k^{\mathrm{cl}} \leq \eta_k^{\mathrm{si}} \leq \eta_k^*. \tag{14}$$

## 3.1 Nonconvex Functions

We are now ready to present a convergence rate result for nonconvex functions.

**Theorem 3.1.** *Let $f$ be an $(L_0, L_1)$-smooth function, and let $\{x_k\}$ be iterate sequence of GM (8) with one of the stepsize choices given by (9), (12) or (13). Then, $\min_{0 \leq k \leq K} \|\nabla f(x_k)\| \leq \epsilon$ for any given $\epsilon > 0$ whenever*

$$K + 1 \geq \frac{2L_0 F_0}{a\epsilon^2} + \frac{3L_1 F_0}{a\epsilon},$$

*where $a = 1$ for stepsizes (9) and (12), and $a = \frac{1}{2}$ for stepsize (13).*

The proof of Theorem 3.1 can be found in Section B.2. The rate in Theorem 3.1 matches, up to absolute constants, the rate in (Koloskova et al., 2023) for clipped GM with $\eta = \frac{1}{9}(L_0 + cL_1)$ for $c = \frac{L_0}{L_1}$, or equivalently the GM with stepsize $\eta_k = \frac{1}{18L_0} \min\{1, \frac{L_0}{L_1\|\nabla f(x_k)\|}\}$. Furthermore, our rate is significantly better than the rate $\mathcal{O}(\frac{L_0 F_0}{\epsilon^2} + \frac{L_1^2 F_0}{L_0})$ obtained in (Zhang et al., 2019) for the clipped GM since $\frac{L_1 F_0}{\epsilon} \leq \frac{L_0^2 F_0}{2\epsilon} + \frac{L_1^2 F_0}{2L_0}$, and the latter expression can be arbitrarily far away from the former whenever $L_0$ is sufficiently small and $L_1$ is distinct from zero. In addition to that, our convergence rate result does not depend on the gradient norm at the initial point, in contrast to Li et al. (2023) who consider a wider class of generalized-smooth functions but whose rate (polynomially) depends on $\|\nabla f(x_0)\|$. Also, our rate from Theorem 3.1 is better than $\mathcal{O}(\frac{L_0 F_0}{\epsilon^2} + \frac{L_1^2 F_0^2}{\epsilon^2})$ provided in (Hübler et al., 2024) for the GM equipped with a certain backtracking line search.

## 3.2 CONVEX FUNCTIONS

Let us now provide the convergence rate for convex functions.

**Theorem 3.2.** *Let $\{x_k\}$ be the iterates of GM* (8) *with one of the stepsize choices given in* (9) (12) *or* (13)*, as applied to problem* (1) *with an $(L_0, L_1)$-smooth convex function $f$. Let $x^*$ be an arbitrary solution to the problem and let $F_0 := f(x_0) - f^*$. Then, the sequence $R_k := \|x_k - x^*\|$, $k \geq 0$, is nonincreasing, and $f(x_K) - f^* \leq \epsilon$ for any given $0 < \epsilon \leq F_0$ whenever*

$$K \geq \frac{2}{a} \frac{L_0 R^2}{\epsilon} + \frac{3}{a} L_1 R \ln \frac{F_0}{\epsilon} \qquad \left( \leq \frac{2 + \frac{3}{e}}{a} \frac{L_0 R^2}{\epsilon} + \frac{3(1 + \frac{1}{e})}{a} [L_1 R]^2 \right),$$

*where $R := R_0$, and $a = 1$ for stepsizes* (9)*,* (12) *and $a = \frac{1}{2}$ for stepsize* (13)*.*

The proof of Theorem 3.2 can be found in Section B.3. Notice, that the second estimate $\mathcal{O}(\frac{L_0 R^2}{\epsilon} + [L_1 R]^2)$ in Theorem 3.2 comes from a very pessimistic bound on $F_0$ with the exponentially large quantity $\exp(L_1 R)\frac{L_0 R^2}{2}$ coming from Lemmas 2.2 and 2.3. However, in the case when $F_0$ is reasonably bounded (e.g., we apply "hot-start" or $f$ is a well-behaved function such as the logistic one), the $\mathcal{O}(L_1 R \ln \frac{F_0}{\epsilon})$ term from the main estimate can be much smaller than $\mathcal{O}([L_1 R]^2)$ from the pessimistic estimate. It is worth mentioning the work of Lobanov et al. (2024), posted online after the ICLR rebuttal, where the same bound as in Theorem 3.2 was independently derived.

In Theorem 3.2, we do not make an assumption on $L$-smoothness of the objective, in contrast to (Koloskova et al., 2023). Moreover, the rate in the theorem is better than $\mathcal{O}(\frac{L_0 R^2}{\epsilon} + \sqrt{\frac{L}{\epsilon}} L_1 R^2)$ provided in (Koloskova et al., 2023) for the clipped GM. Also, in contrast to (Li et al., 2023), our result does not include the gradient norm at the initial point which could be quite large (consider, e.g., $f(x) = \frac{1}{p} \|x\|^p$ from Example A.1 for $p > 2$ and $x_0$ sufficiently far from the origin).

## 4 NORMALIZED GRADIENT METHOD

To run GM from Section 3, it is necessary to know the parameters $(L_0, L_1)$ in advance. In many real-life examples, those parameters are unknown, and it might be computationally expensive to estimate them. Furthermore, for any given function $f$, the pair $(L_0, L_1)$ is generally not unique (see Examples A.1 and A.2), and it is not clear in advance which pair would result in the best possible convergence rate of our optimization method. To address this issue, in this section, we present another version of the gradient method that does not require knowing $(L_0, L_1)$. This is the *normalized gradient method* (NGM):

$$x_{k+1} = x_k - \frac{\beta_k}{\|\nabla f(x_k)\|} \nabla f(x_k), \qquad k \geq 0, \tag{15}$$

where $x_0 \in \mathbb{R}^d$ is a certain starting point, and $\beta_k$ are positive coefficients. The following result describes the efficiency of NGM (see Section C for the proof).

**Theorem 4.1.** *Let $\{x_k\}$ be the iterates of NGM* (15)*, as applied to problem* (1) *with an $(L_0, L_1)$-smooth convex function $f$. Consider the constant coefficients $\beta_k = \frac{\hat{R}}{\sqrt{K+1}}$, $0 \leq k \leq K - 1$, where $\hat{R} > 0$ is a parameter and $K \geq 1$ is the total number of iterations of the method (fixed in advance). Then, $\min_{0 \leq k \leq K} f(x_k) - f^* \leq \epsilon$ for any given $\epsilon > 0$ whenever*

$$K + 1 \geq \max\left\{ \frac{L_0 \bar{R}^2}{\epsilon}, \frac{4}{9} [L_1 \bar{R}]^2 \right\},$$

*where $\bar{R} := \frac{1}{2}(\frac{R^2}{\hat{R}} + \hat{R})$, $R := \|x_0 - x^*\|$, and $x^*$ is an arbitrary solution of the problem.*

The parameter $\hat{R}$ in the formula for coefficients $\beta_k$ is an estimation of the initial distance $R$ to a solution, and the best complexity bound of $K^* := \mathcal{O}(\frac{L_0 R^2}{\epsilon} + [L_1 R]^2)$ is achieved whenever $\hat{R} = R$. Note that, even if $\hat{R} \neq R$, the method still converges but with a slightly worse total complexity of $K^* \rho^2$, where $\rho = \max\{\frac{R}{\hat{R}}, \frac{\hat{R}}{R}\}$.

The proof of Theorem 4.1 is based on the following two important facts (Nesterov, 2018, Section 3). First, under the proper choice of coefficients $\beta_k$, NGM ensures that the minimal value $v_K^*$ among $v_k := \frac{\langle \nabla f(x_k), x_k - x^* \rangle}{\|\nabla f(x_k)\|}$, $0 \le k \le K$, converges to zero at the rate of $\frac{\bar{R}}{\sqrt{K}}$. These quantities $v_k$ have a geometrical meaning—each of them is exactly the distance from the point $x^*$ to the supporting hyperplane to the sublevel set of $f$ at the point $x_k$. Second, whenever $v_K^*$ converges to zero, so does $\min_{0 \le k \le K} f(x_k) - f^*$. Moreover, we can relate the two quantities whenever we can bound, for any given $v \ge 0$, the function residual $f(x) - f^*$ over the ball $\|x - x^*\| \le v$:

**Lemma 4.2** ((Nesterov, 2018, Lemma 3.2.1)). *Let $f \colon \mathbb{R}^d \to \mathbb{R}$ be a differentiable convex function. Then, for any $x, y \in \mathbb{R}^d$ and*[5] $v_f(x; y) := \frac{[\langle \nabla f(x), x-y \rangle]_+}{\|\nabla f(x)\|}$, *it holds that*

$$f(x) - f(y) \le \max_{z \in \mathbb{R}^d} \{f(z) - f(y) : \|z - y\| \le v_f(x; y)\}. \tag{16}$$

In our case—when the function $f$ is $(L_0, L_1)$-smooth—the corresponding bound can be obtained from Lemma 2.2.

In Theorem 4.1, we fix the number of iterations $K$ before running the method, which is a standard approach for the (normalized)-(sub)gradient methods (Section 3.2 in Nesterov (2018)). However, doing so may be undesirable in practice since it becomes difficult to continue running the method if the time budget was suddenly increased and also prevents the method from using larger stepsizes at the initial iterations. To overcome these drawbacks, one can use time-varying coefficients by setting $\beta_k = \frac{\hat{R}}{\sqrt{k+1}}$, $0 \le k \le K - 1$. This results in the same worst-case theoretical complexity as in Theorem 4.1 but with an extra logarithmic factor (see Theorem C.2). Moreover, one can completely eliminate this extra logarithmic factor by switching to an appropriate modification of the standard (sub)gradient method such as Dual Averaging (Nesterov, 2005).

For $\hat{R} = R$, the complexity of NGM is $\mathcal{O}(\frac{L_0 R^2}{\epsilon} + [L_1 R]^2)$ which is generally worse than that of the previously considered GM (see Theorem 3.2 and the corresponding discussion). However, recall that GM requires knowing $(L_0, L_1)$, and its rate depends on the particular choice of these constants. In contrast, NGM does not require the knowledge of these parameters, and its "real" complexity is

$$\mathcal{O}(1) \min_{L_0, L_1} \left\{ \frac{L_0 \bar{R}^2}{\epsilon} + [L_1 \bar{R}]^2 : f \text{ is } (L_0, L_1)\text{-smooth} \right\},$$

where $\mathcal{O}(1)$ is an absolute constant.

## 5 GRADIENT METHOD WITH POLYAK STEPSIZES

In the previous sections, the parameters required to run the methods were $(L_0, L_1)$ for GM, and the estimation $\hat{R}$ of the initial distance to a solution $R$ for NGM. To achieve good complexity for NGM, the estimate $\hat{R}$ should be close to the real $R$, otherwise the algorithm will be inefficient. Sometimes, $(L_0, L_1)$, or a good estimate $\hat{R}$ are unknown, while the optimal value of the objective is available. One such example is overparametrized models in machine learning where $f^* = 0$.

In this section, we focus on the case when $f^*$ is known and analyze the gradient method (8) with the Polyak stepsizes (PS-GM):

$$\eta_k = \frac{f(x_k) - f^*}{\|\nabla f(x_k)\|^2}, \qquad k \ge 0. \tag{17}$$

**Theorem 5.1.** *Let $\{x_k\}$ be the iterates of PS-GM* (8)*,* (17)*, as applied to problem* (1) *with an $(L_0, L_1)$-smooth convex function $f$. Then, it holds that $\min_{0 \le k \le K} f(x_k) - f^* \le \epsilon$ for any given $\epsilon > 0$ whenever*

$$K + 1 \ge \max\left\{ \frac{4L_0 R^2}{\epsilon}, [6L_1 R]^2 \right\},$$

*where $R := \|x_0 - x^*\|$ and $x^*$ is an arbitrary solution of the problem.*

---

[5]Here $[t]_+ := \max\{t, 0\}$ is the nonnegative part of $t \in \mathbb{R}$.

---

**Algorithm 1** AGMsDR

---

1: **Input:** Initial point $x_0 \in \mathbb{R}^d$, update rule $T(\cdot)$.
2: $v_0 = x_0$, $A_0 = 0$, $\zeta_0(x) = \frac{1}{2}\|x - v_0\|^2$.
3: **for** $k = 0, 1, \ldots$ **do**
4: $\quad y_k = \arg\min_y \{f(y) \colon y = v_k + \beta(x_k - v_k), \beta \in [0, 1]\}$.
5: $\quad x_{k+1} = T(y_k)$, $M_k = \frac{\|\nabla f(y_k)\|^2}{2[f(y_k) - f(x_{k+1})]}$ $(> 0)$.[6]
6: $\quad$ Find $a_{k+1} > 0$ from the equation $M_k a_{k+1}^2 = A_k + a_{k+1}$. Set $A_{k+1} = A_k + a_{k+1}$.
7: $\quad v_{k+1} = \arg\min_{x \in \mathbb{R}^d} \{\zeta_{k+1}(x) := \zeta_k(x) + a_{k+1}[f(y_k) + \langle \nabla f(y_k), x - y_k \rangle]\}$.

---

We prove the theorem by using a standard inequality for the gradient method with Polyak stepsizes (PS-GM) for convex functions,

$$R_k^2 - R_{k+1}^2 \geq \frac{f_k^2}{g_k^2},$$

where $R_k = \|x_k - x^*\|$, $f_k = f(x_k) - f^*$, and $g_k = \|\nabla f(x_k)\|$. We then leverage the lower bound (7), and bound the gradient norm $g_k$ by $\psi^{-1}(f_k)$, where $\psi(g) := \frac{g^2}{2L_0 + 3L_1 g}$, obtaining

$$R_k^2 - R_{k+1}^2 \geq \frac{f_k^2}{[\psi^{-1}(f_k)]^2}.$$

Summing up these relations, passing to the minimal value of $f_k$, and rearranging the resulting inequality, we obtain the desired bound. The complete proof of Theorem 5.1 can be found Section D.1.

Notice that the rate $\mathcal{O}(\frac{L_0 R^2}{\epsilon} + [L_1 R]^2)$ in Theorem 5.1 is the same as that of NGM from Theorem 4.1. Further, our rate is better than $\mathcal{O}(\frac{L_0 R^2}{\epsilon} + \sqrt{\frac{L}{\epsilon}} L_1 R^2)$ provided in (Takezawa et al., 2024), and does not require any extra assumptions such as the $L$-Lipschitz smoothness of the objective. Finally, as for NGM, the rate for PS-GM holds for any choice of $(L_0, L_1)$, including the best possible one.

# 6 ACCELERATED GRADIENT METHOD

This section develops an accelerated method for minimizing an $(L_0, L_1)$-smooth convex function $f$. The key ingredient of our analysis is a monotone variant of the accelerated gradient scheme known as the Accelerated Gradient Method with Small-Dimensional Relaxation (AGMsDR) (Nesterov et al., 2021). We present this method in Algorithm 1 in a slightly more general form than the original work. Specifically, instead of computing $x_{k+1}$ via a standard gradient step from $y_k$, we allow any update rule $T(\cdot) \colon \mathbb{R}^d \to \mathbb{R}^d$ that ensures a strictly positive decrease in the function value:

$$f(x) - f(T(x)) > 0, \qquad \forall x \in \mathbb{R}^d \setminus \{x \colon f(x) = f^*\}. \tag{18}$$

**Theorem 6.1.** *Let AGMsDR (Algorithm 1) be applied to problem* (1) *with a differentiable convex objective $f$, and any update rule $T(\cdot)$ satisfying the strictly positive decrease property* (18)*. Let $x^*$ be an arbitrary solution of the problem, and let $R := \|x_0 - x^*\|$. Then, for all $k \geq 0$, we have*

$$f(x_{k+1}) - f^* \leq \frac{2R^2}{\left(\sum_{i=0}^k \frac{1}{\sqrt{M_i}}\right)^2}, \qquad f(x_{k+1}) + \frac{1}{2M_k}\|\nabla f(y_k)\|^2 = f(y_k) \leq f(x_k). \tag{19}$$

The proof of Theorem 6.1 is given in Section E.1. Interestingly, neither the result nor the algorithm assumes any specific smoothness properties of the objective function. However, the convergence rate depends on the magnitude of the quantities $M_i \equiv \frac{\|\nabla f(y_i)\|^2}{2[f(y_i) - f(x_{i+1})]}$, which quantifies the progress made by each step $T(\cdot)$. For standard $L$-Lipschitz smooth functions, a natural choice of $T(\cdot)$ is a gradient step with stepsize $\frac{1}{L}$, yielding $M_i \leq L$ and the well-known rate $\mathcal{O}(\frac{LR^2}{k^2})$ for $f(x_k) - f^*$.

---

[6]For the sake of simplicity, in what follows, we always assume that, at each iteration, $\nabla f(y_k) \neq 0$. Otherwise, $y_k$ is an optimal point, and we can stop the method. Note that, in view of (18), the denominator in the definition of $M_k$ is strictly positive.

For $(L_0, L_1)$-smooth functions, we define $x_{k+1} = T(y_k)$ as a gradient step with any of the stepsize rules discussed in Section 3 (applied to $y_k$ rather than $x_k$), ensuring the following progress per step:

$$f(y_k) - f(x_{k+1}) \geq \frac{g_k^2}{L_0' + L_1' g_k}, \tag{20}$$

where $g_k := \|\nabla f(y_k)\|$, $L_0' := \frac{2}{a} L_0$, and $L_1' := \frac{3}{a} L_1$, with $a$ being an absolute constant depending on the specific stepsize rule. This implies $M_k \leq \frac{1}{2}(L_0' + L_1' g_k)$, leading to the following convergence rate estimate:

$$f(x_{k+1}) - f^* \leq \frac{(2R)^2}{\left(\sum_{i=0}^{k} \frac{1}{\sqrt{L_0' + L_1' g_i}}\right)^2}. \tag{21}$$

To obtain an explicit complexity bound from (21), we must show that the gradient norms $g_i$ do not grow too quickly on average. This follows from (20) and the algorithm's construction ensuring that $f(y_k) \leq f(x_k)$. Ultimately, this yields the following complexity result whose proof is given in Section E.2.

**Theorem 6.2.** *Let AGMsDR (Algorithm 1) be applied to solving problem* (1) *with an $(L_0, L_1)$-smooth convex objective, and $T(\cdot)$ being the gradient update $T(x) = x - \eta_x \nabla f(x)$, where $\eta_x$ is any of the stepsizes* (9), (12), *or* (13) *(with $x_k$ replaced by $x$, respectively). Further, let $x^*$ be an arbitrary solution to the problem, and define $F_0 := f(x_0) - f^*$ and $R := \|x_0 - x^*\|$. Then, $f(x_k) - f^* \leq \epsilon$ for a given $0 < \epsilon \leq F_0$ whenever*

$$k \geq \sqrt{\frac{48 L_0 R^2}{a \epsilon}} + \left\lceil 3(\tfrac{2}{a} L_1 R)^{2/3} \right\rceil \left\lceil \log_2 \frac{2 F_0}{\epsilon} \right\rceil,$$

*where $a = 1$ for stepsize rules* (9), (12), *and $a = \frac{1}{2}$ for stepsize rule* (13). *The total number of first-order oracle queries required to construct $x_k$ is at most $(\nu + 1)k$, where $\nu$ is the number of oracle queries needed to compute $y_k$ at each iteration.*

Compared to existing complexity results for accelerated gradient methods on $(L_0, L_1)$-smooth functions—such as the $\mathcal{O}\left(\left(L_1^2 R^2 + \frac{L_1^2 F_0}{L_0} + 1\right)\sqrt{\frac{F_0 + L_0 R^2}{\epsilon}}\right)$ bound for NAG (Li et al., 2023) (see Section F), and the $\mathcal{O}(1) \exp(\mathcal{O}(1) L_1 R)\sqrt{\frac{L_0 R^2}{\epsilon}}$ bound for STM-Max (Gorbunov et al., 2024)—our complexity estimate in Theorem 6.2 is significantly better.

At each step, AGMsDR requires solving a certain one-dimensional subproblem to compute $y_k$, which we assume requires at most $\nu$ oracle queries. For many practical problems, this subproblem is computationally efficient, making the extra factor $\nu$ in the complexity estimate negligible. Nevertheless, from a theoretical perspective, eliminating this one-dimensional search (as in the standard FGM for Lipschitz-smooth functions) remains an important open question for future research.

## 7 CONCLUSION

This work investigates gradient methods for $(L_0, L_1)$-smooth optimization problems. We have provided new insights into this function class, presented examples, and identified operations preserving $(L_0, L_1)$-smoothness. Additionally, we have established refined properties of these functions, leading to tighter approximations of the objective and its gradient. Building on these improved properties, we have derived new stepsizes for the gradient method and connected them to normalized and clipped stepsizes. For these stepsizes, we have achieved the best-known complexity $\mathcal{O}\left(\frac{L_0 F_0}{\epsilon^2} + \frac{L_1 F_0}{\epsilon}\right)$ for finding an $\epsilon$-stationary point in nonconvex problems. In the convex setting, our analysis significantly strengthens existing results, yielding the improved complexity $\mathcal{O}\left(\frac{L_0 R^2}{\epsilon} + L_1 R \ln \frac{F_0}{\epsilon}\right)$ for the gradient method with our stepsizes. We have further analyzed the GM-PS and NGM methods, both of which achieve the complexity $\mathcal{O}\left(\frac{L_0 R^2}{\epsilon} + [L_1 R]^2\right)$, a significant improvement over previously known bounds. Notably, these methods automatically adapt to the best possible values of $(L_0, L_1)$.

Finally, we have obtained a fast complexity bound of $\nu \mathcal{O}\left(\sqrt{\frac{L_0 R^2}{\epsilon}} + \lceil (L_1 R)^{2/3} \rceil \lceil \ln \frac{F_0}{\epsilon} \rceil\right)$ for AGMsDR, which provides the best efficiency estimate currently available for minimizing $(L_0, L_1)$-smooth convex functions. An interesting open question is whether line search can be eliminated in

the accelerated method, potentially replacing $\nu$ in the complexity bound with an absolute constant. Additionally, it remains to be seen whether the second term in the complexity bound can be further improved or if it is indeed optimal.

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

# A  MISSING PROOFS IN SECTION 2

## A.1  PROOF OF LEMMA 2.2

(2) $\implies$ (3). Let $x, y \in \mathbb{R}^d$ be arbitrary and let $h := y - x \neq 0$ (otherwise the claim is trivial). Then, for any $t \in [0, 1]$, using (2), we can estimate

$$\|\nabla f(x+th) - \nabla f(x)\| \leq \|h\| \int_0^t \|\nabla^2 f(x+\tau h)\| d\tau \leq \|h\| \int_0^t (L_0 + L_1 \|\nabla f(x+\tau h)\|) d\tau =: \chi(t).$$

Our goal is to upper bound $\chi(1)$. We may assume that $L_1 > 0$ since otherwise $\chi(1) = L_0 \|h\|$ and the proof is finished. Differentiating, we obtain, for any $t \in [0, 1]$,

$$\chi'(t) = L_0 \|h\| + L_1 \|h\| \|\nabla f(x + th)\| \leq (L_0 + L_1 \|\nabla f(x)\|) \|h\| + L_1 \|h\| \chi(t),$$

where the final bound is due to the triangle inequality and the previous display. Hence, for any $t \in [0, 1]$, we have

$$\frac{d}{dt} \ln\left[(L_0 + L_1 \|\nabla f(x)\| + \epsilon) \|h\| + L_1 \|h\| \chi(t)\right] \leq L_1 \|h\|,$$

where $\epsilon > 0$ is arbitrary[7]. Integrating this inequality in $t \in [0, 1]$ and noting that $\chi(0) = 0$, we get

$$\ln \frac{L_0 + L_1 \|\nabla f(x)\| + \epsilon + L_1 \chi(1)}{L_0 + L_1 \|\nabla f(x)\| + \epsilon} \leq L_1 \|h\|,$$

or, equivalently,

$$\chi(1) \leq (L_0 + L_1 \|\nabla f(x)\| + \epsilon) \frac{e^{L_1 \|h\|} - 1}{L_1}.$$

Passing now to the limit as $\epsilon \to 0$, we obtain (3).

[(3) $\implies$ (4)] Let $x, y \in \mathbb{R}^d$ be arbitrary points and let $h := y - x$. Then, using (3), we can estimate

$$|f(y) - f(x) - \langle \nabla f(x), y - x \rangle| \leq \int_0^1 |\langle \nabla f(x + th) - \nabla f(x), h \rangle| dt$$

$$\leq (L_0 + L_1 \|\nabla f(x)\|) \|h\| \int_0^1 \frac{e^{L_1 \|h\| t} - 1}{L_1} dt = (L_0 + L_1 \|\nabla f(x)\|) \frac{e^{L_1 \|h\|} - L_1 \|h\| - 1}{L_1^2},$$

which is exactly (4).

[(4) $\implies$ (2)] Let us fix an arbitrary point $x \in \mathbb{R}^d$ and an arbitrary unit vector $h \in \mathbb{R}^d$. Then, for any $t > 0$, it follows from (4) that

$$|f(x + th) - f(x) - t \langle \nabla f(x), h \rangle| \leq (L_0 + L_1 \|\nabla f(x)\|) \frac{e^{L_1 t} - L_1 t - 1}{L_1^2}.$$

Dividing both sides by $t^2$ and passing to the limit as $t \to 0$, we get

$$|\langle \nabla^2 f(x) h, h \rangle| \leq L_0 + L_1 \|\nabla f(x)\|.$$

This proves (2) since the unit vector $h$ was allowed to be arbitrary. $\square$

## A.2  PROOF OF LEMMA 2.4

*Proof of* (5). Let $x, y \in \mathbb{R}^d$ be arbitrary points and let us assume w.l.o.g. that $L_1 > 0$. In view of the convexity of $f$ and (4), for any $h \in \mathbb{R}^d$, we can write the following two inequalities:

$$\begin{aligned} 0 &\leq f(y + h) - f(x) - \langle \nabla f(x), y + h - x \rangle \\ &\leq \beta_f(x, y) + \langle \nabla f(y) - \nabla f(x), h \rangle + \frac{L_0 + L_1 \|\nabla f(y)\|}{L_1^2} \phi(L_1 \|h\|), \end{aligned}$$

---

[7]This additional term is needed to handle the possibility of $L_0 + L_1 \|\nabla f(x)\|$ being zero.

where $\beta_f(x,y) := f(y) - f(x) - \langle \nabla f(x), y - x \rangle$. Denoting $a := L_0 + L_1\|\nabla f(y)\| > 0$ and $s := \nabla f(y) - \nabla f(x)$, we therefore obtain

$$\beta_f(x,y) \geq \max_{h\in\mathbb{R}^d}\Big\{\langle s,h\rangle - \frac{a}{L_1^2}\phi(L_1\|h\|)\Big\} = \max_{r\geq 0}\Big\{\|s\|r - \frac{a}{L_1^2}\phi(L_1 r)\Big\} = \frac{a}{L_1^2}\phi_*\Big(\frac{L_1\|s\|}{a}\Big).$$

[Proof of (6)] Summing up (5) with the same inequality but $x$ and $y$ interchanged, we obtain (6).

[Proof of (7)] By using a lower bound $\phi_*(\gamma) \geq \frac{\gamma^2}{2+\gamma}$ in (5) and denoting $a = \|\nabla f(x) - \nabla f(y)\|$ and $g = \|\nabla f(y)\|$, we obtain

$$f(y) \geq f(x) + \langle \nabla f(x), y - x \rangle + \frac{L_0 + L_1 g}{L_1^2}\frac{L_1^2 a^2}{(L_0 + L_1 g)^2}\frac{L_0 + L_1 g}{2(L_0 + L_1 g) + a}$$

$$= f(x) + \langle \nabla f(x), y - x \rangle + \frac{a^2}{2(L_0 + L_1 g) + a}. \qquad \square$$

### A.3 EXAMPLES AND PROPERTIES OF $(L_0, L_1)$-SMOOTH FUNCTIONS

Let us present a few simple examples of $(L_0, L_1)$-smooth functions.

*Example* A.1. The function $f(x) = \frac{1}{p}\|x\|^p$, where $p > 2$, is $(L_0, L_1)$-smooth with arbitrary $L_1 > 0$ and $L_0 = (\frac{p-2}{L_1})^{p-2}$.

*Example* A.2. The function $f(x) = \ln(1 + e^x)$ is $(L_0, L_1)$-smooth with arbitrary $L_1 \in [0,1]$ and $L_0 = \frac{1}{4}(1 - L_1)^2$.

The preceding examples also show that the choice of $L_0, L_1$ parameters is generally not unique. While we cannot guarantee that the class is closed under all standard operations, such as the summation, affine substitution of the argument, we can still show that some operations do preserve $(L_0, L_1)$-smoothness under certain additional assumptions.

**Proposition A.3.** *Let $f: \mathbb{R}^d \to \mathbb{R}$ be a twice continuously differentiable $(L_0, L_1)$-smooth function. Then, the following statements hold:*

1. *Let $g: \mathbb{R}^d \to \mathbb{R}$ be an $L$-smooth and $M$-Lipschitz twice continuously differentiable function. Then, the sum $f + g$ is $(L_0', L_1')$-smooth with $L_0' = L_0 + ML_1 + L$ and $L_1' = L_1$.*

2. *Let $f_i: \mathbb{R}^{d_i} \to \mathbb{R}$ be an $(L_{0,i}, L_{1,i})$-smooth function for each $i = 1, \ldots, n$. Then, the function $h: \mathbb{R}^{d_1} \times \ldots \times \mathbb{R}^{d_n} \to \mathbb{R}$ given by $h(x) = \sum_{i=1}^n f_i(x_i)$, where $x = (x_1, \ldots, x_n)$, is $(L_0, L_1)$-smooth with $L_0 = \max_{1\leq i\leq n} L_{0,i}$ and $L_1 = \max_{1\leq i\leq n} L_{1,i}$.*

3. *If $f$ is univariate $(d = 1)$ and $h(x) = f(\langle a, x \rangle + b)$, $x \in \mathbb{R}^d$, where $a \in \mathbb{R}^d$, $b \in \mathbb{R}$, then $h$ is $(L_0', L_1')$-smooth with parameters $L_0' = \|a\|^2 L_0$ and $L_1' = \|a\| L_1$.*

4. *Let additionally $\nabla^2 f(x) \succ 0$ for all $x \in \mathbb{R}^d$ and $f$ be 1-coercive[8]. Then, $f$ is $(L_0, L_1)$-smooth iff its conjugate $f_*$ (which is, under our assumptions, defined on the entire space and also twice continuously differentiable) satisfies $\nabla^2 f_*(s) \succeq \frac{1}{L_0 + L_1\|s\|} I$ for all $s \in \mathbb{R}^d$, where $I$ is the identity matrix.*

One simple example of the additive term $g$ satisfying the assumptions in the first item of Proposition A.3 is an affine function (for which $L = 0$); another interesting example is the soft-max function $g(x) = \mu\ln(\sum_{i=1}^m e^{[\langle a_i, x\rangle + b_i]/\mu})$, where $a_i \in \mathbb{R}^d$, $b_i \in \mathbb{R}$, $\mu > 0$. Based on the second statement of Proposition A.3 and Example A.1, the function $f(x) = \frac{1}{p}\|x\|_p^p \equiv \frac{1}{p}\sum_{i=1}^d |x_i|^p$ with $p > 2$ is $(L_0, L_1)$-smooth with arbitrary $L_1 > 0$ and $L_0 = (\frac{p-2}{L_1})^{p-2}$. Using the third statement, we can generalize Example A.2 and conclude that $f(x) = \ln(1 + e^{\langle a, x\rangle})$ is also $(L_0, L_1)$-smooth with arbitrary $L_1 \in [0, \|a\|]$ and $L_0 = \frac{1}{4}(\|a\| - L_1)^2$. Also, we can use the last statement of the proposition to show that $f(x) = \frac{L_0}{L_1^2}\phi(L_1\|x\|) \equiv \frac{L_0}{L_1^2}(e^{L_1\|x\|} - L_1\|x\| - 1)$ is $(L_0, L_1)$-smooth since the Hessian of its conjugate $f_*(s) = \frac{L_0}{L_1^2}\phi_*(\frac{L_1\|s\|}{L_0}) \equiv \frac{L_0}{L_1^2}[(1 + \frac{L_1\|s\|}{L_0})\ln(1 + \frac{L_1\|s\|}{L_0}) - \frac{L_1\|s\|}{L_0}]$ has the

---

[8]This means that $\frac{f(x)}{\|x\|} \to +\infty$ as $\|x\| \to \infty$.

form $\nabla^2 f_*(s) = \frac{1}{L_0 + L_1\|s\|} I$. In particular, we can construct an $(L_0, L_1)$-smooth function by taking any convex function $h_*$, adding to it $\phi_*$ and taking the conjugate (this corresponds to the infimal convolution of $h$ with $\phi$).

### A.4 PROOF OF EXAMPLE A.1

*Proof.* Differentiating, we obtain, for any $x \in \mathbb{R}^d$,

$$\nabla f(x) = \|x\|^{p-2} x, \qquad \nabla^2 f(x) = \|x\|^{p-2}\left(I + (p-2)\frac{xx^\top}{\|x\|^2}\right),$$

where $I$ is the identity matrix. Hence, for any $L_1 > 0$, the minimal value of $L_0$ satisfying the inequality from Definition 2.1 is given by

$$L_0 = \max_{x \in \mathbb{R}^d}\left\{\|\nabla^2 f(x)\| - L_1\|\nabla f(x)\|\right\} = \max_{x \in \mathbb{R}^d}\left\{(p-1)\|x\|^{p-2} - L_1\|x\|^{p-1}\right\}$$

$$= \max_{\tau \geq 0}\left\{(p-1)\tau^{\frac{p-2}{p-1}} - L_1\tau\right\}.$$

The solution of the latter problem is $\tau^* = (\frac{p-2}{L_1})^{p-1}$. Substituting this value, we obtain

$$L_0 = (p-1)\left(\frac{p-2}{L_1}\right)^{p-2} - L_1\left(\frac{p-2}{L_1}\right)^{p-1} = \left(\frac{p-2}{L_1}\right)^{p-2}. \qquad \square$$

### A.5 PROOF OF EXAMPLE A.2

*Proof.* Differentiating, we obtain, for any $x \in \mathbb{R}$,

$$f'(x) = \frac{e^x}{1 + e^x} \in (0, 1), \quad f''(x) = \frac{e^x}{(1 + e^x)^2} = f'(x)(1 - f'(x)).$$

Thus, for any $L_1 \in [0, 1]$, the minimal value of $L_0$ satisfying the inequality from Definition 2.1 is

$$L_0 = \max_{x \in \mathbb{R}}\{|f''(x)| - L_1|f'(x)|\} = \max_{\tau \in (0,1)}\{\tau(1-\tau) - L_1\tau\}$$

$$= \max_{\tau \in (0,1)}\{(1 - L_1)\tau - \tau^2\} = \frac{1}{4}(1 - L_1)^2. \qquad \square$$

### A.6 PROOF OF PROPOSITION A.3

*Proof.* [Claim 1] Since, $g$ and $\nabla g$ are $M$ and $L$ Lipschitz continuous, $\|\nabla g(x)\| \leq M$ and $\|\nabla^2 g(x)\| \leq L$ for all $x \in \mathbb{R}$. Let $F = f + g$, then, for any $x \in \mathbb{R}^d$, we can estimate

$$\|\nabla^2 F(x)\| \leq \|\nabla^2 f(x)\| + \|\nabla^2 g(x)\| \leq L_0 + L + L_1\|\nabla f(x)\|$$

$$\leq L_0 + L + L_1\|\nabla g(x)\| + L_1\|\nabla F(x)\|$$

$$\leq (L_0 + L_1 M + L) + L_1\|\nabla F(x)\|.$$

[Claim 2] Notice, that the gradient of $f$ is $\nabla f(x) = (\nabla f_1(x_1)^\top, \ldots, \nabla f_n(x_n)^\top)^\top$ and the Hessian of $f$ is $\nabla^2 f(x)$ is a block-diagonal matrix, with $\nabla^2 f_i(x_i)$ blocks. Thus,

$$\|\nabla^2 f(x)\| = \max_{1 \leq i \leq n} \|\nabla^2 f_i(x_i)\| \leq \max_{1 \leq i \leq n} \{L_{0,i} + L_{1,i}\|\nabla f_i(x_i)\|\}$$

$$\leq \max_{1 \leq i \leq n} \{L_{0,i} + L_{1,i}\|\nabla f(x)\|\} \leq \max_{1 \leq i \leq n} L_{0,i} + (\max_{1 \leq i \leq n} L_{1,i})\|\nabla f(x)\|.$$

[Claim 3] Observe that the gradient of a function is $\nabla f(x) = g'(\langle a, x\rangle + b)a$, and the Hessian is $\nabla^2 f(x) = g''(\langle a, x\rangle + b)aa^\top$. Hence,

$$\|\nabla^2 f(x)\| = |g''(\langle a, x\rangle + b)|\|a\|^2 \leq (L_0 + L_1|g'(\langle a, x\rangle + b)|)\|a\|^2$$

$$= L_0\|a\|^2 + \|a\|L_1\|\nabla f(x)\|.$$

[Claim 4] Under our assumptions, $s = \nabla f(x)$ is a one-to-one transformation from $\mathbb{R}^d$ to $\mathbb{R}^d$ (whose inverse transformation is $x = \nabla f_*(s)$); moreover, the Hessians at such a pair of points are inverse to each other: $\nabla^2 f_*(s) = [\nabla^2 f(x)]^{-1}$ (see, e.g., Corollaries 4.1.4 and 4.2.10 in (Hiriart-Urruty & Lemaréchal, 1993), as well as Example 11.9 from (Rockafellar & Wets, 2009)). Thus, for any pair of points $x, s \in \mathbb{R}^d$ such that $s = \nabla f(x)$, our assumption $\|\nabla^2 f(x)\| \leq L_0 + L_1\|\nabla f(x)\|$ which, due to the convexity of $f$, can be equivalently rewritten as $\nabla^2 f(x) \preceq (L_0 + L_1\|\nabla f(x)\|)I$, is equivalent to

$$\nabla^2 f_*(s) \equiv [\nabla^2 f(x)]^{-1} \succeq \frac{1}{L_0 + L_1\|\nabla f(x)\|}I \equiv \frac{1}{L_0 + L_1\|s\|}I.$$

This proves the claim since the transformation $s = \nabla f(x)$ is one-to-one. $\qquad\square$

### A.7 Proof of Lemma 2.3

*Proof.* [Claim 1] Indeed, for any $t \in [0, 3)$, we have

$$\phi(t) = e^t - t - 1 = \sum_{i=2}^{\infty} \frac{t^i}{i!} = \sum_{i=0}^{\infty} \frac{t^{2+i}}{(2+i)!} = \frac{t^2}{2}\sum_{i=0}^{\infty} \frac{t^i}{\prod_{j=3}^{2+i} j} \leq \frac{t^2}{2}\sum_{i=0}^{\infty} \frac{t^i}{3^i} = \frac{t^2}{2(1 - \frac{t}{3})}.$$

Similarly, for any $t \geq 0$,

$$\phi(t) = \frac{t^2}{2}\sum_{i=0}^{\infty} \frac{t^i}{\prod_{j=3}^{2+i} j} \leq \frac{t^2}{2}\sum_{i=0}^{\infty} \frac{t^i}{i!} = \frac{t^2}{2}e^t.$$

[Claim 2] By the definition, for any $\gamma \geq 0$, we have

$$\phi_*(\gamma) = \max_{t \geq 0}\{\gamma t - \phi(t)\} = \max_{t \geq 0}\{(1 + \gamma)t - e^t\} + 1.$$

Differentiating, we see that the solution of this optimization problem is $t_* = \ln(1 + \gamma)$. Hence,

$$\phi_*(\gamma) = (1 + \gamma)\ln(1 + \gamma) - (1 + \gamma) + 1 = (1 + \gamma)\ln(1 + \gamma) - \gamma.$$

[Claim 3] We first show that, for any $\gamma \geq 0$,

$$\ln(1 + \gamma) \geq \frac{2\gamma}{2 + \gamma}.$$

Since both functions coincide at $\gamma = 0$, it suffices to verify the corresponding inequality for the derivatives:

$$\frac{1}{1 + \gamma} \geq \frac{4}{(2 + \gamma)^2} \equiv \frac{4}{4 + 4\gamma + \gamma^2} \equiv \frac{1}{1 + \gamma + \frac{\gamma^2}{4}}.$$

But this is obviously true. Applying the derived inequality, we get, for any $\gamma \geq 0$,

$$\phi_*(\gamma) \equiv (1 + \gamma)\ln(1 + \gamma) - \gamma \geq \frac{2\gamma(1 + \gamma)}{2 + \gamma} - \gamma = \frac{\gamma[2(1 + \gamma) - (2 + \gamma)]}{2 + \gamma} = \frac{\gamma^2}{2 + \gamma},$$

which proves the first part of the claim.

For the second part, we note that $\phi_*(\gamma)$ and $\frac{\gamma^2}{2}$ coincide at $\gamma = 0$. Hence, it suffices to check the corresponding inequality for the derivatives, i.e., to verify that, for all $\gamma \geq 0$,

$$\phi'_*(\gamma) \equiv \ln(1 + \gamma) \leq \gamma.$$

But this follows from the concavity of the logarithm. $\qquad\square$

## B Missing Proofs in Section 3

### B.1 One-Step Progress

**Lemma B.1.** *Let $f : \mathbb{R}^d \to \mathbb{R}$ be an $(L_0, L_1)$-smooth function, let $x \in \mathbb{R}^d$, and let $T(x) = x - \eta\nabla f(x)$, where $\eta$ is given by one of the following formulas:*

$$(1)\ \eta_* = \frac{1}{L_1\|\nabla f(x)\|}\ln\Big(1 + \frac{L_1\|\nabla f(x)\|}{L_0 + L_1\|\nabla f(x)\|}\Big), \quad (2)\ \eta_{\mathrm{si}} = \frac{1}{L_0 + \frac{3}{2}L_1\|\nabla f(x)\|},$$

$$(3)\ \eta_{\mathrm{cl}} = \min\Big\{\frac{1}{2L_0}, \frac{1}{3L_1\|\nabla f(x)\|}\Big\}.$$

*Then,*

$$f(x) - f(T(x)) \geq \frac{a\|\nabla f(x)\|^2}{2L_0 + 3L_1\|\nabla f(x)\|},$$

*where $a = 1$ in cases (1) and (2), and $a = \frac{1}{2}$ in case (3).*

*Proof.* [Case (1)] The proof of this case was already presented in Section 3.

For the other two cases, we start by applying Lemma 2.2 to get

$$\Delta := f(x) - f(T(x)) \geq \langle \nabla f(x), x - T(x) \rangle - \frac{L_0 + L_1\|\nabla f(x)\|}{L_1^2} \phi(L_1\|T(x) - x\|)$$

$$= \eta_* g^2 - \frac{L_0 + L_1 g}{L_1^2} \phi(\eta_* L_1 g),$$

where $g := \|\nabla f(x)\|$ and $\phi(t) = e^t - t - 1$.

[Case (2)] Estimating $\phi(t) \leq \frac{3t^2}{6-2t} \leq \frac{t^2}{2-t}$ (Lemma 2.3) and substituting the definition of $\eta_{\mathrm{si}}$, we can continue as follows:

$$\Delta \geq \eta_{\mathrm{si}} g^2 - \frac{L_0 + L_1 g}{L_1^2} \frac{\eta_{\mathrm{si}}^2 L_1^2 g^2}{2 - \eta_{\mathrm{si}} L_1 g} = \left(1 - \frac{(L_0 + L_1 g)\eta_{\mathrm{si}}}{2 - \eta_{\mathrm{si}} L_1 g}\right) \eta_{\mathrm{si}} g^2$$

$$= \left(1 - \frac{L_0 + L_1 g}{(L_0 + \frac{3}{2}L_1 g)(2 - \frac{L_1 g}{L_0 + \frac{3}{2}L_1 g})}\right) \frac{g^2}{L_0 + \frac{3}{2}L_1 g} = \frac{g^2}{2L_0 + 3L_1 g}.$$

[Case (3)] Observe that

$$\frac{1}{2L_0 + 3L_1 g} \leq \eta_{\mathrm{cl}} \equiv \frac{1}{\max\{2L_0, 3L_1 g\}} \leq \frac{1}{L_0 + \frac{3}{2}L_1 g}.$$

Combining these bounds with $\phi(t) \leq \frac{3t^2}{6-2t}$ (Lemma 2.3 (1)), we get

$$\Delta \geq \eta_{\mathrm{cl}} g^2 - \frac{L_0 + L_1 g}{L_1^2} \frac{3L_1^2 \eta_{\mathrm{cl}}^2 g^2}{6 - 2\eta_{\mathrm{cl}} L_1 g} = \left(1 - \frac{3\eta_{\mathrm{cl}}(L_0 + L_1 g)}{6 - 2\eta_{\mathrm{cl}} L_1 g}\right) \eta_{\mathrm{cl}} g^2$$

$$\geq \left(1 - \frac{3(L_0 + L_1 g)}{(L_0 + \frac{3}{2}L_1 g)(6 - \frac{2L_1 g}{L_0 + \frac{3}{2}L_1 g})}\right) \frac{g^2}{2L_0 + 3L_1 g}$$

$$= \left(1 - \frac{3(L_0 + L_1 g)}{6L_0 + 7L_1 g}\right) \frac{g^2}{2L_0 + 3L_1 g} \geq \frac{1}{2} \frac{g^2}{2L_0 + 3L_1 g}. \qquad \square$$

**Lemma B.2.** *Let $f \colon \mathbb{R}^d \to \mathbb{R}$ be a convex $(L_0, L_1)$-smooth function, let $x \in \mathbb{R}^d$, and let $T(\cdot)$ be any of the update rules from Lemma B.1. Further, let $x^*$ be a minimizer of $f$. Then,*

$$\|T(x) - x^*\| \leq \|x - x^*\|.$$

*Proof.* Denote $\beta = \langle \nabla f(x), x - x^* \rangle$ and $g = \|\nabla f(x)\|$. According to the update rule $T(\cdot)$, we have

$$\|T(x) - x^*\| = \|x - x^*\|^2 - 2\eta\beta + \eta^2 g^2.$$

Therefore, to prove that $\|T(x) - x^*\| \leq \|x - x^*\|^2$, we need to show that

$$\eta g^2 \leq 2\beta.$$

Applying bound (7) twice, we see that

$$\beta_k \equiv [f(x) - f^*] + [f^* - f(x) - \langle \nabla f(x), x^* - x \rangle]$$

$$\geq \frac{g^2}{2L_0 + 3L_1 g} + \frac{g^2}{2L_0 + L_1 g} \geq \frac{g^2}{L_0 + L_1 g},$$

where the final inequality follows from the fact that $\frac{1}{a} + \frac{1}{b} \geq \frac{4}{a+b}$ (convexity of $t \mapsto \frac{1}{t}$). Thus, we need to check if

$$\eta \leq \frac{2}{L_0 + L_1 g}. \tag{22}$$

Furthermore, it suffices to check this inequality only for the largest among the three stepsizes we consider. This is the stepsize $\eta^*$ (see (14)). Applying $\ln(1 + \gamma) \le \gamma$ (which holds for any $\gamma \ge 0$), we see that

$$\eta^* \equiv \frac{1}{L_1 g} \ln\left(1 + \frac{L_1 g}{L_0 + L_1 g}\right) \le \frac{1}{L_0 + L_1 g},$$

so (22) is indeed satisfied. □

### B.2 PROOF OF THEOREM 3.1

*Proof.* According to Lemma B.1, for any $k \ge 0$, we have

$$f(x_k) - f(x_{k+1}) \ge \frac{a\|\nabla f(x_k)\|^2}{2L_0 + 3L_1\|\nabla f(x_k)\|},$$

where $a$ is an absolute constant defined in the statement depending on the stepsize choice. Denote $f_k = f(x_k) - f^* (\ge 0)$ and $g_k = \|\nabla f(x_k)\|$. In this notation, the above inequality reads

$$f_k - f_{k+1} \ge a\psi(g_k), \qquad \psi(\gamma) := \frac{\gamma^2}{2L_0 + 3L_1\gamma}.$$

Summing up these inequalities for all $0 \le k \le K$ and denoting $g_K^* = \min_{0 \le k \le K} g_k$, we get

$$F_0 \ge f_0 - f_K \ge a \sum_{k=0}^{K} \psi(g_k) \ge a(K+1)\psi(g_K^*),$$

where the final inequality holds since $\psi$ is an increasing function. Denoting the corresponding inverse function by $\psi^{-1}$, we come to the conclusion that

$$g_K^* \le \psi^{-1}\left(\frac{F_0}{a(K+1)}\right) \le \epsilon$$

whenever

$$\frac{F_0}{a(K+1)} \le \psi(\epsilon),$$

or, equivalently,

$$K + 1 \ge \frac{F_0}{a\psi(\epsilon)} \equiv \frac{2L_0 F_0}{a\epsilon^2} + \frac{3L_1 F_0}{a\epsilon}. \qquad \square$$

### B.3 PROOF OF THEOREM 3.2

*Proof of Theorem 3.2.* Let $k \ge 0$ be arbitrary and denote $f_k := f(x_k) - f^*$ and $g_k := \|\nabla f(x_k)\|$. According to Lemma B.1, we have

$$f_k - f_{k+1} \ge a\psi(g_k), \qquad \psi(\gamma) := \frac{\gamma^2}{2L_0 + 3L_1\gamma},$$

where $a$ is an absolute constant defined in the statement depending on the stepsize choice. Further, according to Lemma B.2, the distances $R_k := \|x_k - x^*\|$ are nonincreasing. In particular, $R_k \le R_0 \equiv R$. Hence, in view of the convexity of $f$, we can estimate

$$f_k \le \langle \nabla f(x_k), x_k - x^* \rangle \le g_k R_k \le g_k R.$$

Combining the above two displays and using the fact that the function $\psi$ is increasing, we obtain

$$f_k - f_{k+1} \ge a\psi\left(\frac{f_k}{R}\right).$$

Consequently,

$$a \le \frac{f_k - f_{k+1}}{\psi(\frac{f_k}{R})} \le \int_{f_{k+1}}^{f_k} \frac{dt}{\psi(\frac{t}{R})} = \int_{f_{k+1}}^{f_k} \left(\frac{2L_0 R^2}{t^2} + \frac{3L_1 R}{t}\right) dt$$

$$= 2L_0 R^2 \left(\frac{1}{f_{k+1}} - \frac{1}{f_k}\right) + 3L_1 R \ln\frac{f_k}{f_{k+1}}.$$

Summing up these inequalities for all $0 \leq k \leq K - 1$ and dropping the negative $\frac{1}{f_0}$ term, we get

$$aK \leq \frac{2L_0 R^2}{f_K} + 3L_1 R \ln \frac{f_0}{f_K}.$$

Hence, $f_K \leq \epsilon$ whenever

$$K \geq \frac{2L_0 R^2}{a\epsilon} + \frac{3}{a} L_1 R \ln \frac{f_0}{\epsilon} =: K(\epsilon).$$

To upper bound $K(\epsilon)$, we first estimate $f_0$ using Lemmas 2.2 and 2.3:

$$f_0 \leq \frac{L_0}{L_1^2} \phi(L_1 R) \leq \frac{L_0 R^2}{2} e^{L_1 R}.$$

This gives us

$$aK(\epsilon) \leq \frac{2L_0 R^2}{\epsilon} + 3L_1 R \left( L_1 R + \ln \frac{L_0 R^2}{\epsilon} \right) = \frac{2L_0 R^2}{\epsilon} + 3[L_1 R]^2 + 6L_1 R \ln \left( \sqrt{\frac{L_0 R^2}{\epsilon}} \right).$$

Estimating $\ln t \leq \frac{t}{e}$ (holding for any $t > 0$) and applying the AM-GM inequality, we get

$$aK(\epsilon) \leq \frac{2L_0 R^2}{\epsilon} + 3[L_1 R]^2 + \frac{6}{e} \sqrt{\frac{L_0 R^2}{\epsilon}} [L_1 R]^2 \leq \frac{(2 + \frac{3}{e}) L_0 R^2}{\epsilon} + \left( 3 + \frac{3}{e} \right) [L_1 R]^2. \quad \square$$

## C MISSING PROOFS IN SECTION 4

### C.1 GENERAL RESULT

**Lemma C.1.** *Let $\{x_k\}$ be the iterates of NGM* (15) *with arbitrary coefficients $\beta_k > 0$, as applied to problem* (1) *with an $(L_0, L_1)$-smooth convex function $f$. Then, $\min_{0 \leq k \leq K} f(x_k) - f^* \leq \epsilon$ for any given $K \geq 0$ and $\epsilon > 0$ whenever*

$$\delta_K := \frac{R^2 + \sum_{k=0}^{K} \beta_k^2}{2 \sum_{k=0}^{K} \beta_k} \leq \delta(\epsilon) := \min \left\{ \frac{3}{2L_1}, \sqrt{\frac{\epsilon}{L_0}} \right\},$$

*where $R := \|x_0 - x^*\|$ is the distance from the initial point to a solution $x^*$ of the problem.*

*Proof.* According to (15), for any $k \geq 0$, we have

$$\|x_{k+1} - x^*\|^2 = \|x_k - x^*\|^2 - 2\eta_k \langle \nabla f(x_k), x_k - x^* \rangle + \eta_k^2 \|\nabla f(x_k)\|^2$$
$$= \|x_k - x^*\|^2 - 2\beta_k v_k + \beta_k^2,$$

where $v_k := \frac{\langle \nabla f(x_k), x_k - x^* \rangle}{\|\nabla f(x_k)\|} (\geq 0)$. Summing up these relations over $k = 0, \ldots, K$ and rearranging the terms, we obtain

$$2 \sum_{k=0}^{K} \beta_k v_k \leq R^2 + \sum_{k=0}^{K} \beta_k^2.$$

Denoting $v_K^* = \min_{0 \leq k \leq K} v_k$, we get

$$v_K^* \leq \frac{R^2 + \sum_{k=0}^{K} \beta_k^2}{2 \sum_{k=0}^{K} \beta_k} =: \delta_K. \tag{23}$$

Let $f_K^* := \min_{0 \leq k \leq K} f(x_k)$. Then, by Lemma 4.2,

$$f_K^* - f^* \leq \max_z \{ f(z) - f^* : \|z - x^*\| \leq v_K^* \}.$$

Applying Lemma 2.2 and the fact that $\phi(t) \leq \frac{3t^2}{6 - 2t}$ for any $t \in [0, 3)$ (Lemma 2.3), we obtain

$$f_K^* - f^* \leq \frac{L_0}{L_1^2} \phi(L_1 v_K^*) \leq \frac{3L_0 (v_K^*)^2}{6 - 2L_1 v_K^*}$$

whenever $L_1 v_k^* < 3$. To achieve the desired accuracy $\epsilon$, it thus suffices to ensure that the following two inequalities are satisfied:

$$2L_1 v_K^* \leq 3, \qquad L_0(v_K^*)^2 \leq \epsilon.$$

This is equivalent to

$$v_K^* \leq \min\Big\{\frac{3}{2L_1}, \sqrt{\frac{\epsilon}{L_0}}\Big\} =: \delta(\epsilon),$$

and follows from $\delta_k \leq \delta(\epsilon)$ in view of (23). □

## C.2 Proof of Theorem 4.1

*Proof.* According to Lemma C.1, we need to ensure that

$$\delta_K := \frac{R^2 + \sum_{k=0}^{K} \beta_k^2}{2 \sum_{k=0}^{K} \beta_k} \leq \delta(\epsilon) := \min\Big\{\frac{3}{2L_1}, \sqrt{\frac{\epsilon}{L_0}}\Big\}.$$

In our case,

$$\delta_K = \frac{R^2 + \hat{R}^2}{2\hat{R}\sqrt{K+1}} = \frac{\bar{R}}{\sqrt{K+1}}.$$

Therefore, $\delta_K \leq \delta(\epsilon)$ iff

$$K + 1 \geq \frac{\bar{R}^2}{\delta^2(\epsilon)} \equiv \max\Big\{\frac{4}{9}[L_1\bar{R}]^2, \frac{L_0\bar{R}^2}{\epsilon}\Big\}. \qquad □$$

## C.3 Analysis for Time-Varying Step Size

**Theorem C.2.** *Let $\{x_k\}$ be the iterates of NGM (15), as applied to problem (1) with an $(L_0, L_1)$-smooth nonlinear[9] convex function $f$. Consider decreasing coefficients $\beta_k = \frac{\hat{R}}{\sqrt{k+1}}$, $k \geq 0$, where $\hat{R} > 0$ is a parameter. Then, $\min_{0 \leq k \leq K} f(x_k) - f^* \leq \epsilon$ for any given $\epsilon > 0$ whenever*

$$K + 1 \geq \max\Big\{4N_{\bar{R}}(\epsilon), \Big(\frac{e}{e-1}\Big)^2 N_{\hat{R}}(\epsilon)[\ln(4N_{\hat{R}}(\epsilon))]_+^2\Big\},$$

*where $\bar{R} := \frac{1}{2}(\frac{R^2}{\hat{R}} + \hat{R})$, $R := \|x_0 - x^*\|$ ($x^*$ is an arbitrary solution of the problem), and*

$$N_D(\epsilon) := \max\Big\{\frac{4}{9}[L_1 D]^2, \frac{L_0 D^2}{\epsilon}\Big\}.$$

*Proof.* According to Lemma C.1, we need to ensure that

$$\delta_K := \frac{R^2 + \sum_{k=0}^{K} \beta_k^2}{2 \sum_{k=0}^{K} \beta_k} \leq \delta(\epsilon) := \min\Big\{\frac{3}{2L_1}, \sqrt{\frac{\epsilon}{L_0}}\Big\}.$$

For our choice of $\beta_k$, we obtain, by standard results (e.g., Lemma 2.6.3 in Rodomanov (2022)), that

$$\sum_{k=0}^{K} \beta_k^2 = \hat{R}^2 \sum_{k=1}^{K+1} \frac{1}{k} \leq \hat{R}^2[1 + \ln(K+1)], \qquad \sum_{k=0}^{K} \beta_k = \hat{R} \sum_{k=1}^{K+1} \frac{1}{\sqrt{k}} \geq \hat{R}\sqrt{K+1}.$$

Hence,

$$\delta_K \leq \frac{R^2 + \hat{R}^2[1 + \ln(K+1)]}{2\hat{R}\sqrt{K+1}} = \frac{\bar{R}}{\sqrt{K+1}} + \frac{\hat{R}\ln(K+1)}{2\sqrt{K+1}}.$$

To ensure that $\delta_K \leq \delta(\epsilon)$, it suffices to ensure that the following two inequalities are satisfied:

$$\frac{\bar{R}}{\sqrt{K+1}} \leq \frac{\delta(\epsilon)}{2}, \qquad \frac{\hat{R}\ln(K+1)}{\sqrt{K+1}} \leq \delta(\epsilon).$$

---

[9]This means that $L_0 + L_1\|\nabla f(x)\| > 0$ for any $x \in \mathbb{R}^d$, see Lemma 2.2.

The first inequality is equivalent to $K+1 \geq \frac{4\bar{R}^2}{\delta^2}$. To get the second one, it suffices to take, according to Lemma C.3 (with $p = \frac{1}{2}$ and $\delta' = \frac{\delta(\epsilon)}{\hat{R}}$),

$$K + 1 \geq \left(\frac{e}{e-1}\frac{2\hat{R}}{\delta(\epsilon)}\left[\ln\frac{2\hat{R}}{\delta(\epsilon)}\right]_+\right)^2 \equiv \left(\frac{e}{e-1}\right)^2\frac{\hat{R}^2}{\delta^2(\epsilon)}\left[\ln\frac{4\hat{R}^2}{\delta^2(\epsilon)}\right]_+^2.$$

Putting these two inequalities together and substituting our formula for $\delta(\epsilon)$, we come to the requirement that

$$K + 1 \geq \max\left\{\frac{4\bar{R}^2}{\delta^2(\epsilon)}, \left(\frac{e}{e-1}\right)^2\frac{\hat{R}^2}{\delta^2(\epsilon)}\left[\ln\frac{4\hat{R}^2}{\delta^2(\epsilon)}\right]_+^2\right\}$$

$$= \max\left\{4N_{\bar{R}}(\epsilon), \left(\frac{e}{e-1}\right)^2 N_{\hat{R}}(\epsilon)[\ln(4N_{\hat{R}}(\epsilon))]_+^2\right\},$$

where

$$N_D(\epsilon) := \frac{D^2}{\delta^2(\epsilon)} = \max\left\{\frac{4}{9}[L_1D]^2, \frac{L_0D^2}{\epsilon}\right\}. \qquad \square$$

**Lemma C.3.** *For any real $p, \delta > 0$, we have the following implication[10]:*

$$t \geq \left(\frac{e}{e-1}\frac{[\ln\frac{1}{p\delta}]_+}{p\delta}\right)^{\frac{1}{p}} \qquad \Longrightarrow \qquad \frac{\ln t}{t^p} \leq \delta.$$

*Proof.* W.l.o.g., we can assume that $p = 1$, and our goal is to prove the implication

$$t \geq \frac{e}{e-1}\frac{[\ln\frac{1}{\delta}]_+}{\delta} =: t(\delta) \qquad \Longrightarrow \qquad \phi(t) := \frac{\ln t}{t} \leq \delta.$$

The general case then follows by the change of variables $t = (t')^p$ and $\delta = p\delta'$.

Further, we can assume that $\delta \leq \frac{1}{e}$ since otherwise $\phi(t) \leq \frac{1}{e} \leq \delta$ for any $t \geq 0$ (since the maximum of $\phi$ is achieved at $t_* = e$). Under this additional assumption, $[\ln\frac{1}{\delta}]_+ = \ln\frac{1}{\delta}$.

Let us now assume that $t \geq t(\delta)$ ($\geq \frac{e^2}{e-1} \geq e$ since $\delta \leq \frac{1}{e}$). Since the function $\phi$ is decreasing on the interval $[e, +\infty)$, we have

$$\phi(t) \leq \phi(t(\delta)) = \frac{\ln t(\delta)}{t(\delta)} = \frac{\ln t(\delta)}{\frac{e}{e-1}\ln\frac{1}{\delta}}\delta.$$

To finish the proof, it remains to show that the final fraction in the above display is $\leq 1$, or, equivalently, that

$$t(\delta) \equiv \frac{e}{e-1}\frac{\ln\frac{1}{\delta}}{\delta} \leq \left(\frac{1}{\delta}\right)^{\frac{e}{e-1}}.$$

Rearranging and denoting $u := \left(\frac{1}{\delta}\right)^{\frac{1}{e-1}}$, we see that the above inequality is equivalent to

$$\phi(u) \equiv \frac{\ln u}{u} \leq \frac{1}{e}.$$

But this is indeed true since $\phi$ attains its maximum value at $u = e$. $\qquad \square$

## D  MISSING PROOFS IN SECTION 5

### D.1  PROOF OF THEOREM 5.1

*Proof.* Let $x^*$ be an arbitrary solution. By the method's update rule and convexity of $f(\cdot)$, we get, for all $k \geq 0$,

$$\|x_{k+1} - x^*\|^2 = \|x_k - x^*\|^2 - 2\eta_k\langle\nabla f(x_k), x_k - x^*\rangle + \eta_k^2\|\nabla f(x_k)\|^2$$

$$\leq \|x_k - x^*\|^2 - \frac{[f(x_k) - f^*]^2}{\|\nabla f(x_k)\|^2}.$$

---

[10]For $t = 0$, we define by continuity $\frac{\ln t}{t^p} \equiv -\infty$.

Denote $R_k = \|x_k - x^*\|$, $g_k = \|\nabla f(x_k)\|$ and $f_k = f(x_k) - f^*$. According to Lemma 2.4, for each $k \geq 0$, it holds that

$$f_k \geq \psi(g_k), \qquad \text{where} \quad \psi(g) := \frac{g^2}{2L_0 + 3L_1 g}, \quad g \geq 0.$$

Observe that the function $\psi$ is increasing, so its inverse $\psi^{-1}$ is well-defined and is increasing as well. In terms of this function, $g_k \leq \psi^{-1}(f_k)$ and hence

$$R_k^2 - R_{k+1}^2 \geq \frac{f_k^2}{g_k^2} \geq \Big(\frac{f_k}{\psi^{-1}(f_k)}\Big)^2.$$

Summing up these inequalities over $0 \leq k \leq K$ and rearranging, we get

$$\sum_{k=0}^{K} \Big(\frac{f_k}{\psi^{-1}(f_k)}\Big)^2 \leq R_0^2 - R_{K+1}^2 \leq R_0^2 \equiv R^2.$$

Note that $\frac{\psi^{-1}(t)}{t}$ is increasing in $t$ (as the composition of increasing in $\gamma$ function $\frac{\psi(\gamma)}{\gamma} \equiv \frac{\gamma}{2L_0 + 3L_1\gamma}$ with increasing in $t$ function $\gamma = \psi^{-1}(t)$). Thus, by taking a minimum over the terms on the left-hand side of the above display and denoting $f_K^* := \min_{0 \leq k \leq K} f_k$, we get

$$(K+1)\Big(\frac{f_K^*}{\psi^{-1}(f_K^*)}\Big)^2 \leq R^2.$$

Rearranging, we obtain

$$\psi^{-1}(f_K^*) \geq \frac{\sqrt{K+1} f_K^*}{R},$$

or, equivalently,

$$f_K^* \geq \psi\Big(\frac{\sqrt{K+1} f_K^*}{R}\Big) \equiv \frac{(K+1)(f_K^*)^2}{R^2\big(2L_0 + 3L_1\frac{\sqrt{K+1}f_K^*}{R}\big)} = \frac{(f_K^*)^2}{\frac{2L_0 R^2}{K+1} + \frac{3L_1 R}{\sqrt{K+1}} f_K^*}.$$

Hence,

$$f_K^* \leq \frac{2L_0 R^2}{(K+1)(1 - 3L_1 R\sqrt{K+1})},$$

whenever $3L_1 R\sqrt{K+1} < 1$. Thus, to achieve desired accuracy $\epsilon > 0$, the number $K$ of iterations should satisfy the following conditions:

$$3L_1 R\sqrt{K+1} \leq \frac{1}{2}, \qquad \frac{4L_0 R^2}{K+1} \leq \epsilon.$$

Thus, the final iteration complexity is $K+1 \geq \max\{\frac{4L_0 R^2}{\epsilon}, [6L_1 R]^2\}$. $\qquad \square$

# E  MISSING PROOFS IN SECTION 6

The proof of Theorem 6.1 is similar to the original proof Theorem 1 in (Nesterov et al., 2021), but, instead of the Lipschitz-smoothness of $f$, we use the definition of $M_k$.

## E.1  PROOF OF THEOREM 6.1

*Proof.* Let us prove by induction that, for any $k \geq 0$, we have

$$A_k f(x_k) \leq \zeta_k^* := \zeta_k(v_k). \tag{24}$$

This trivially holds for $k = 0$ since $A_0 = 0$ and $\zeta_0^* = 0$. Now assume that (24) is satisfied for some $k \geq 0$ and let us prove that it is also satisfied for the next index $k' = k + 1$. We start by noting that

$$\zeta_{k+1}^* = \zeta_{k+1}(v_{k+1}) = \zeta_k(v_{k+1}) + a_{k+1}[f(y_k) + \langle \nabla f(y_k), v_{k+1} - y_k \rangle]$$

$$\geq \zeta_k^* + \frac{1}{2}\|v_{k+1} - v_k\|^2 + a_{k+1}[f(y_k) + \langle \nabla f(y_k), v_{k+1} - y_k \rangle]$$

$$\geq A_k f(x_k) + \frac{1}{2}\|v_{k+1} - v_k\|^2 + a_{k+1}[f(y_k) + \langle \nabla f(y_k), v_{k+1} - y_k \rangle], \tag{25}$$

where the first inequality holds due to the strong convexity of $\zeta_k$, and the second one is due to the induction hypothesis. Further, note that, by construction, $y_k \in [v_k, x_k]$. Considering separately any of the three possible situations, $y_k = v_k$, $y_k = x_k$ and $y_k \in (v_k, x_k)$, we see that, in all cases,

$$\langle \nabla f(y_k), v_k - y_k \rangle \geq 0.$$

Substituting this estimate into (25) and using the fact that $f(y_k) \leq f(x_k)$ (by construction), we obtain

$$\zeta_{k+1}^* \geq A_k f(x_k) + a_{k+1} f(y_k) + \frac{1}{2} \|v_{k+1} - v_k\|^2 + a_{k+1} \langle \nabla f(y_k), v_{k+1} - v_k \rangle$$

$$\geq A_{k+1} f(y_k) + \frac{1}{2} \|v_{k+1} - v_k\|^2 + a_{k+1} \langle \nabla f(y_k), v_{k+1} - v_k \rangle$$

$$\geq A_{k+1} f(y_k) - \frac{a_{k+1}^2}{2} \|\nabla f(y_k)\|^2 = A_{k+1} \left[ f(y_k) - \frac{1}{2M_k} \|\nabla f(y_k)\|^2 \right] = A_{k+1} f(x_{k+1}),$$

where the final identity is due to the definition of $M_k$, while the preceeding one follows from the definition of $a_{k+1}$, which ensures that

$$M_k a_{k+1}^2 = A_{k+1}. \tag{26}$$

The induction is now complete.

Let $k \geq 1$ be arbitrary. By the convexity of $f$ and the definition of $A_k$, we have

$$\zeta_k^* \leq \zeta_k(x^*) = \frac{1}{2} R^2 + \sum_{i=0}^{k-1} a_{i+1} [f(y_i) + \langle \nabla f(y_i), x^* - y_i \rangle] \leq \frac{1}{2} R^2 + A_k f^*.$$

where $R \equiv \|x_0 - x^*\|$. Combining this with (24), we conclude that

$$f(x_k) - f^* \leq \frac{R^2}{2A_k}. \tag{27}$$

It remains to estimate the rate of growth of the coefficients $A_k$. From (26) and the definition of $A_{k+1}$, it follows, for any $k \geq 0$, that

$$\sqrt{\frac{A_{k+1}}{M_k}} = a_{k+1} = A_{k+1} - A_k = (\sqrt{A_{k+1}} + \sqrt{A_k})(\sqrt{A_{k+1}} - \sqrt{A_k})$$

$$\leq 2\sqrt{A_{k+1}}(\sqrt{A_{k+1}} - \sqrt{A_k}).$$

Cancelling $\sqrt{A_{k+1}}$ on both sides and telescoping the resulting inequalities, we get, for any $k \geq 1$,

$$A_k \geq \frac{1}{4} \left( \sum_{i=0}^{k-1} \sqrt{\frac{1}{M_i}} \right)^2.$$

Substituting this estimate into (27), we obtain the first relation in (19). The second one follows trivially from the definition of $M_k$ and the fact that $f(y_k) \leq f(x_k)$. $\qquad \square$

### E.2 PROOF OF THEOREM 6.2

*Proof.* Let $k \geq 0$ be arbitrary, and denote $f_k := f(x_k) - f^*$ and $g_k := \|\nabla f(y_k)\|$. According to Theorem 6.1, we have

$$f_{k+1} \leq \frac{2R^2}{\left( \sum_{i=0}^k \frac{1}{\sqrt{M_i}} \right)^2}, \qquad f_k - f_{k+1} \geq \frac{g_k^2}{2M_k},$$

where $M_k = \frac{\|\nabla f(y_k)\|^2}{2[f(y_k) - f(x_{k+1})]}$. Further, from the fact that $x_{k+1} = T(y_k)$ and Lemma B.1, we know that

$$M_k \leq \frac{2L_0 + 3L_1 g_k}{2a} \equiv \frac{1}{2}(L_0' + L_1' g_k),$$

where $L_0' := \frac{2}{a} L_0$, $L_1' := \frac{3}{a} L_1$, and $a$ is as defined in the statement. Thus,

$$f_{k+1} \le \frac{(R')^2}{\left(\sum_{i=0}^{k} \frac{1}{\sqrt{L_0'+L_1'g_i}}\right)^2}, \qquad f_k - f_{k+1} \ge \frac{g_k^2}{L_0' + L_1' g_k},$$

where $R' := 2R$. Applying Lemma E.1, we conclude that $f_k \le \epsilon$ for a given $0 < \epsilon \le f_0$ whenever $k \ge K(\epsilon)$, where

$$
\begin{aligned}
K(\epsilon) &:= \sqrt{\frac{6L_0'(R')^2}{\epsilon}} + \left\lceil 3^{1/3}(L_1' R')^{2/3} \right\rceil \left\lceil \log_2 \frac{2f_0}{\epsilon} \right\rceil \\
&= \sqrt{\frac{6(\frac{2}{a}L_0)(2R)^2}{\epsilon}} + \left\lceil 3^{1/3}\{(\frac{3}{a}L_1)(2R)\}^{2/3} \right\rceil \left\lceil \log_2 \frac{2f_0}{\epsilon} \right\rceil \\
&= \sqrt{\frac{48 L_0 R^2}{a\epsilon}} + \left\lceil 3(\frac{2}{a}L_1 R)^{2/3} \right\rceil \left\lceil \log_2 \frac{2f_0}{\epsilon} \right\rceil.
\end{aligned}
$$

To estimate the oracle complexity, it remains to note that each iteration of the algorithm requires exactly one computation of the gradient plus at most $\nu$ oracle queries for the line search. Hence, the overall oracle complexity to compute $x_k$ is at most $(\nu + 1)k$. $\qquad\square$

**Lemma E.1.** *Let $(f_k)_{k=0}^{\infty}$, $(g_k)_{k=0}^{\infty}$ be nonnegative real sequences such that, for any $k \ge 0$, the following inequalities hold:*

$$f_{k+1} \le \frac{R^2}{\left(\sum_{i=0}^{k} \frac{1}{\sqrt{L_0+L_1 g_i}}\right)^2}, \qquad f_k - f_{k+1} \ge \frac{g_k^2}{L_0 + L_1 g_k},$$

*where $R, L_0, L_1 \ge 0$ are certain constants, and $L_0 + L_1 g_k > 0$ for all $k \ge 0$. Then, for any integer $k \ge 0$ and $N \ge 1$, it holds that*

$$f_{k+N} \le \frac{3 L_0 R^2}{2N^2} + \frac{3(L_1 R)^2}{2N^3} f_k.$$

*Consequently, $f_k \le \epsilon$ for a given $0 < \epsilon \le f_0$ whenever*

$$k \ge \sqrt{\frac{6 L_0 R^2}{\epsilon}} + \left\lceil 3^{1/3}(L_1 R)^{2/3} \right\rceil \left\lceil \log_2 \frac{2f_0}{\epsilon} \right\rceil.$$

*Proof.* Let $k \ge 0$ and $N \ge 1$ be arbitrary. Denote $\bar{g}_{k,N} := \frac{1}{N} \sum_{i=k}^{k+N-1} g_i$. Then, dropping part of the nonnegative terms and applying Jensen's inequality to the convex function $\tau \mapsto \frac{1}{\sqrt{\tau}}$, we see that

$$\sum_{i=0}^{k+N-1} \frac{1}{\sqrt{L_0 + L_1 g_i}} \ge \sum_{i=k}^{k+N-1} \frac{1}{\sqrt{L_0 + L_1 g_i}} \ge \frac{N}{\sqrt{L_0 + L_1 \bar{g}_{k,N}}}.$$

Hence,

$$f_{k+N} \le \frac{(L_0 + L_1 \bar{g}_{k,N}) R^2}{N^2}.$$

Our goal now is to estimate how fast $\bar{g}_{k,N}$ can grow.

According to our assumptions, for any $i \ge 0$, we have $f_i - f_{i+1} \ge \psi(g_i)$, where $\psi\colon [0, +\infty) \to \mathbb{R}$ is an increasing convex function $\psi(g) := \frac{g^2}{L_0 + L_1 g}$. Summing up these inequalities and applying Jensen's inequality, we obtain

$$f_k - f_{k+N} \ge \sum_{i=k}^{k+N-1} \psi(g_i) \ge N\psi(\bar{g}_{k,N}) \quad (\ge 0).$$

Hence, $\bar{g}_{k,N} \le \psi^{-1}\left(\frac{f_k - f_{k+N}}{N}\right)$, where $\psi^{-1}$ is the inverse function of $\psi$. Consequently,

$$f_{k+N} \le \frac{[L_0 + L_1 \psi^{-1}\left(\frac{f_k - f_{k+N}}{N}\right)] R^2}{N^2}.$$

Note that, for any $\gamma \geq 0$, we have $\psi^{-1}(\gamma) = \sqrt{L_0\gamma + \frac{1}{4}L_1^2\gamma^2} + \frac{1}{2}L_1\gamma \leq \sqrt{L_0\gamma} + L_1\gamma$, whence

$$L_0 + L_1\psi^{-1}(\gamma) \leq L_0 + L_1(\sqrt{L_0\gamma} + L_1\gamma) = L_0 + L_1^2\gamma + \sqrt{L_0L_1^2\gamma} \leq \frac{3}{2}(L_0 + L_1^2\gamma).$$

Thus,

$$f_{k+N} \leq \frac{3(L_0 + L_1^2\frac{f_k - f_{k+N}}{N})R^2}{2N^2} = \frac{3L_0R^2}{2N^2} + \frac{3(L_1R)^2}{2N^3}(f_k - f_{k+N}),$$

which proves the first part of the claim.

Applying now Lemma E.2, we conclude that $f_k \leq \epsilon$ for a given $0 < \epsilon \leq f_0$ whenever

$$k \geq \sqrt{\frac{4 \cdot \frac{3}{2}L_0R^2}{\epsilon}} + \left\lceil (2 \cdot \frac{3}{2}(L_1R)^2)^{1/3} \right\rceil \left\lceil \log_2 \frac{2f_0}{\epsilon} \right\rceil$$

$$= \sqrt{\frac{6L_0R^2}{\epsilon}} + \left\lceil 3^{1/3}(L_1R)^{2/3} \right\rceil \left\lceil \log_2 \frac{2f_0}{\epsilon} \right\rceil. \qquad \square$$

**Lemma E.2.** *Let $(f_k)_{k=0}^{\infty}$ be a nonnegative sequence of reals such that, for any integer $k \geq 0$ and $N \geq 1$, it holds that*

$$f_{k+N} \leq \frac{\alpha}{N^2} + \frac{\beta}{N^3}f_k,$$

*where $\alpha, \beta \geq 0$ are certain constants. Then, $f_k \leq \epsilon$ for a given $0 < \epsilon \leq f_0$ whenever*

$$k \geq \sqrt{\frac{4\alpha}{\epsilon}} + \left\lceil (2\beta)^{1/3} \right\rceil \left\lceil \log_2 \frac{2f_0}{\epsilon} \right\rceil.$$

*Proof.* We assume that $\beta > 0$ (otherwise the claim is trivial). Let $N_1 := \lceil (2\beta)^{1/3} \rceil$ ($\geq 1$). Then, for any $k \geq 0$, we have

$$f_{k+N_1} \leq \frac{\alpha}{N_1^2} + \frac{\beta}{N_1^3}f_k \leq \Delta + \frac{1}{2}f_k,$$

where $\Delta := \frac{\alpha}{N_1^2} \leq \frac{\alpha}{(2\beta)^{2/3}}$. Applying now Lemma E.3 to the subsequence $(f_{N_1t})_{t=0}^{\infty}$, we obtain, for any $t \geq 0$, that

$$f_{N_1t} \leq 2\Delta + \frac{1}{2^t}f_0.$$

Hence, for any $t \geq 0$ and any $N \geq 1$, it holds that

$$f_{N_1t+N} \leq \frac{\alpha}{N^2} + \frac{\beta}{N^3}f_{N_1t} \leq \frac{\alpha}{N^2} + \frac{2\beta\Delta}{N^3} + \frac{\beta f_0}{N^32^t} \leq \frac{\alpha}{N^2}\left(1 + \frac{N_1}{N}\right) + \frac{\beta f_0}{N^32^t}.$$

Therefore, to ensure that $f_{N_1t+N} \leq \epsilon$, it suffices to satisfy the following three inequalities:

$$\frac{2\alpha}{N^2} \leq \frac{\epsilon}{2}, \qquad N \geq N_1, \qquad \frac{\beta f_0}{N^32^t} \leq \frac{\epsilon}{2}.$$

Note that, for each $N \geq N_1$, we have $\frac{\beta}{N^3} \leq \frac{\beta}{N_1^3} \leq \frac{1}{2}$. Hence, to satisfy the above three inequalities, it suffices to ensure that

$$N \geq \max\left\{\sqrt{\frac{4\alpha}{\epsilon}}, N_1\right\} =: N_2, \qquad t \geq T := \left\lceil \log_2 \frac{f_0}{\epsilon} \right\rceil.$$

We have thus proved that $f_k \leq \epsilon$ whenever $k \geq N_1T + N_2$. It remains to note that

$$N_1T + N_2 \leq N_1(T+1) + \sqrt{\frac{4\alpha}{\epsilon}} = \left\lceil (2\beta)^{1/3} \right\rceil \left\lceil \log_2 \frac{2f_0}{\epsilon} \right\rceil + \sqrt{\frac{4\alpha}{\epsilon}},$$

where we have first estimated the maximum by the sum and then used the fact that $\lceil \log_2 \tau \rceil + 1 = \lceil \log_2 \tau + 1 \rceil = \lceil \log_2(2\tau) \rceil$ for any $\tau \geq 1$. $\qquad \square$

**Lemma E.3.** *Let $(\gamma_k)_{k=0}^{\infty}$ be a nonnegative real sequence such that, for any $k \geq 0$,*

$$\gamma_{k+1} \leq \Delta + q\gamma_k,$$

*where $\Delta \geq 0$ and $q \in [0, 1)$ are certain constants. Then, for any $k \geq 1$, it holds that*

$$\gamma_k \leq \frac{1-q^k}{1-q}\Delta + q^k\gamma_0 \leq \frac{\Delta}{1-q} + q^k\gamma_0.$$

*Proof.* We can assume that $q > 0$ since otherwise the claim is trivial. Dividing both sides of the inequality from the statement by $q^{k+1}$, we obtain, for any $k \geq 0$,

$$\frac{\gamma_{k+1}}{q^{k+1}} \leq \frac{\Delta}{q^{k+1}} + \frac{\gamma_k}{q^k}.$$

Summing up these inequalities, we get, for any $k \geq 1$,

$$\frac{\gamma_k}{q^k} \leq \sum_{i=0}^{k-1} \frac{\Delta}{q^{i+1}} + \frac{\gamma_0}{q^0} = \Delta \sum_{i=1}^{k} \frac{1}{q^i} + \gamma_0 = \frac{1}{q}\frac{\frac{1}{q^k}-1}{\frac{1}{q}-1}\Delta + \gamma_0 = \frac{\frac{1}{q^k}-1}{1-q}\Delta + \gamma_0,$$

and the claim follows. $\qquad\square$

## F COMPLEXITY OF NAG

Unfortunately, the NAG algorithm presented in (Li et al., 2023) is not scale-invariant and its complexity reported in (Li et al., 2023, Theorem 4.4) is not written explicitly. To streamline the comparison of the complexity bound for NAG with those for other methods for minimizing an $(L_0, L_1)$-smooth function, we provide a simple fix making the algorithm scale-invariant and also rewrite the result of (Li et al., 2023, Theorem 4.4) (assuming it is true) in an explicit form.

**Theorem F.1.** *Consider problem (1) with an $(L_0, L_1)$-smooth convex function $f$ assuming $L_0 > 0$. Let NAG (Li et al., 2023) be applied to solving the rescaled version of this problem:*

$$\tilde{f}^* := \min_{x \in \mathbb{R}^d}\left\{\tilde{f}(x) := \frac{1}{L_0}f(x)\right\},$$

*starting from a certain point $x_0 \in \mathbb{R}^d$. Then, for an appropriate choice of parameters, NAG finds a point $\bar{x} \in \mathbb{R}^d$ such that $f(\bar{x}) - f^* \leq \epsilon$ for a given $\epsilon > 0$ after at most the following number of iterations / gradient-oracle queries:*

$$16\left(128L_1^2R^2 + \frac{128L_1^2F_0}{L_0} + 1\right)\sqrt{\frac{F_0 + L_0R^2}{\epsilon}},$$

*where $F_0 := f(x_0) - f^*$, $R := \|x_0 - x^*\|$ and $x^*$ is an arbitrary solution of our problem.*

*Proof.* By construction, $\tilde{f}$ is an $(\tilde{L}_0, \tilde{L}_1)$-smooth with $\tilde{L}_0 = 1$ and $\tilde{L}_1 = L_1$. In the terminology of (Li et al., 2023), this means that $\tilde{f}$ is $\ell$-smooth w.r.t. the function

$$\ell(G) := \tilde{L}_0 + \tilde{L}_1 G \equiv 1 + L_1 G.$$

Theorem 4.4 from (Li et al., 2023) then tells us that the sequence of the iterates $\{x_t\}$ constructed by NAG satisfies

$$\tilde{f}(x_t) - \tilde{f}^* \leq \frac{4(\tilde{F}_0 + R^2)}{\eta t^2 + 4}, \tag{28}$$

where $\tilde{F}_0 := \tilde{f}(x_0) - \tilde{f}^*$, $R := \|x_0 - x^*\|$, and $\eta > 0$ is the stepsize parameter required to satisfy

$$\eta \leq \min\left\{\frac{1}{16[\ell(2G)]^2}, \frac{1}{2\ell(2G)}\right\} \equiv \frac{1}{16[\ell(2G)]^2} \equiv \frac{1}{16(1 + 2L_1G)^2}, \tag{29}$$

where $G$ is an arbitrary constant such that

$$G \geq \max\{8\sqrt{\ell(2G)(\tilde{F}_0 + R^2)}, \tilde{g}_0\} \equiv \max\{8\sqrt{(1 + 2L_1G)(\tilde{F}_0 + R^2)}, \tilde{g}_0\}. \tag{30}$$

where $\tilde{g}_0 := \|\nabla \tilde{f}(x_0)\|$.

In terms of our original function $f$, the guarantee (28) reads

$$f_t := f(x_t) - f^* \leq \frac{4(F_0 + L_0 R^2)}{\eta t^2 + 4}.$$

To achieve the fastest possible convergence, we select the largest possible stepsize $\eta$ which is, according to (29),

$$\eta = \frac{1}{16(1 + 2L_1 G)^2}.$$

Substituting this formula into the previous display and dropping the (useless for improving the convergence rate) constant 4 from the denominator, we obtain

$$f_t \leq \frac{64(1 + 2L_1 G)^2 (F_0 + L_0 R^2)}{t^2} \leq \epsilon$$

whenever

$$t \geq 8(1 + 2L_1 G)\sqrt{\frac{F_0 + L_0 R^2}{\epsilon}} =: t(\epsilon). \tag{31}$$

The obtained $t(\epsilon)$ is exactly the iteration complexity of the algorithm for obtaining an $\epsilon$-approximate solution for the original problem, and is also its gradient oracle complexity since the method makes precisely one gradient-oracle query at each iteration.

It remains to choose the smallest possible parameter $G$ satisfying (30). We start with rewriting this inequality in terms of the original function:

$$G \geq \max\left\{8\sqrt{(1 + 2L_1 G)\left(\frac{F_0}{L_0} + R^2\right)}, \frac{g_0}{L_0}\right\} \equiv \max\left\{\sqrt{(1 + 2L_1 G)\Delta}, \frac{g_0}{L_0}\right\}$$

where $g_0 := \|\nabla f(x_0)\|$ and $\Delta := 64(\frac{F_0}{L_0} + R^2)$. This inequality is equivalent to the system of two inequalities:

$$G^2 \geq (1 + 2L_1 G)\Delta, \qquad G \geq \frac{g_0}{L_0}.$$

Rearranging, we see that the first inequality is equivalent to

$$G \geq \sqrt{\Delta + L_1^2 \Delta^2} + L_1 \Delta =: G_*$$

Further, it turns out that $G_* \geq \frac{g_0}{L_0}$. Indeed, according to (7), we have $F_0 \geq \frac{g_0^2}{2L_0 + 3L_1 g_0}$, meaning that $g_0 \leq \sqrt{2L_0 F_0 + \frac{9}{4}L_1^2 F_0^2} + \frac{3}{2}L_1 F_0 \leq \sqrt{2L_0 F_0} + 3L_1 F_0$; on the other hand, estimating $\Delta \geq \frac{64F_0}{L_0}$, we see that $L_0(\sqrt{\Delta} + L_1 \Delta) \geq 8\sqrt{L_0 F_0} + 64L_1 F_0$. Thus, the smallest possible value of $G$ satisfying the original requirement (30) is in fact $G = G_*$.

Choosing now $G = G_*$ and substituting the definition of $\Delta$, we obtain

$$1 + 2L_1 G = \frac{G_*^2}{\Delta} \leq \frac{2(\Delta + L_1^2 \Delta^2) + 2L_1^2 \Delta^2}{\Delta} = 2(1 + 2L_1^2 \Delta) = 2\left(1 + \frac{128L_1^2 F_0}{L_0} + 128L_1^2 R^2\right).$$

Substituting this bound into (31), we obtain the claimed bound on $t(\epsilon)$. $\qquad\square$

## G NUMERICAL RESULTS

In Fig. 1, we compare the performance of the analyzed methods for solving optimization problem (1) with a function $f(x) = \frac{1}{p}\|x\|^p$. We fix $L_1 = 1$ and choose $L_0 = (\frac{p-2}{L_1})^{p-2}$ according to Example A.1. For GM, we choose stepsizes according to (9), (12) and (13). For NGM, we use time-varying coefficients $\beta_k = \frac{\hat{R}}{k+1}$ with different values of $\hat{R} \in \{\frac{1}{2}R, 2R, 10R\}$, which allows us to study the robustness of this method to our initial guess of the unknown initial distance to the solution. Note that, for this particular problem, the choice of $\hat{R} = R$ is rather special and allows the method to find the exact solution after one iteration, so we are not considering it. We observe that,

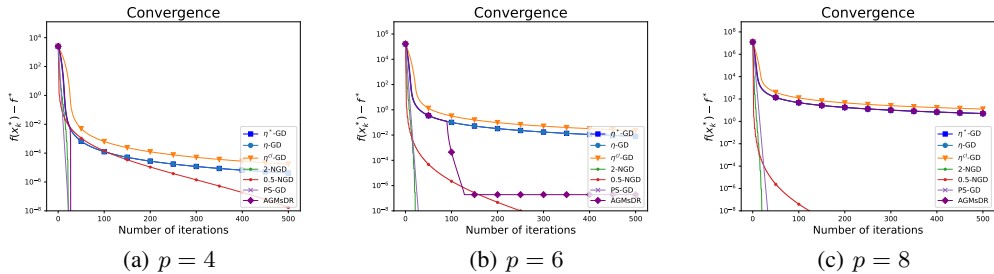

(a) $p = 4$        (b) $p = 6$        (c) $p = 8$

Figure 1: Comparison of gradient methods for $f(x) = \frac{1}{p}\|x\|^p$. $\frac{\hat{R}}{R}$-NGD stands for Normalized Gradient Method, where $\hat{R}$ is an estimation of the true initial distance to a solution $R$. $\eta_*$-GD, $\eta^{\text{si}}$-GD, $\eta^{\text{cl}}$-GD stand for gradient method with stepsizes (9), (12) and (13) respectively, PS-GD stands for Polyak stepsizes gradient method, and AGMsDR stands for Algorithm 1.

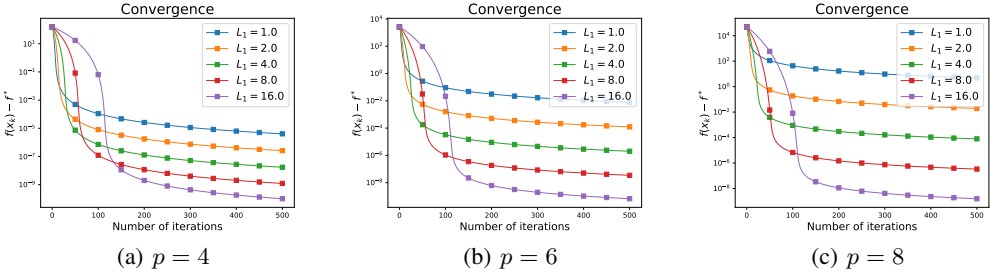

(a) $p = 4$        (b) $p = 6$        (c) $p = 8$

Figure 2: Convergence of the gradient method on the same function but with different choices of $(L_0, L_1)$.

NGM and PS-GM outperform GM with stepsizes from (9), (12) and (13). This can be explained by the fact that the complexity of GM depends on the particular choice of $(L_0, L_1)$, while complexity of NGM and PS-GM involves the optimal parameters $L_0, L_1$ as discussed in Section 4. Moreover, closer initial distance estimation $\hat{R}$ to a true value $R$ leads to a faster convergence of NGM to a solution.

In Fig. 2, we present an experiment studying the performance of the GM with the stepsize rule (9) based on the choice of $(L_0, L_1)$. For each choice of $L_1 \in \{1, 2, 4, 8, 16\}$ we set $L_0 = (\frac{p-2}{L_1})^{p-2}$, according to Example A.1. As expected from the theory (see the corresponding discussion at the end of Section 4), the choice of $(L_0, L_1)$ pair is crucial in practice for the performance of GM and depends on a target accuracy $\epsilon$.

In Fig. 3, we conduct an experiment for accelerated methods and consider GM with stepsize (9), Algorithm 1 with $T(\cdot)$ being the gradient update with stepsize (9), and two variants of normalized Similar Triangles Methods (STM, and STM-Max) from Gorbunov et al. (2024). STM uses normalization by the norm of the gradient at the current point in a gradient step, while STM-Max normalizes by the largest norm of the gradient over the optimization trajectory. It is worth noticing that only STM-Max has theoretical convergence guarantees. We set $L_1 = 1$, $L_0 = (\frac{p-2}{L_1})^{p-2}$ (see Example A.1) with various $p$. We observe that for smaller values of $p$, Algorithm 1 outperforms STM and STM-max.

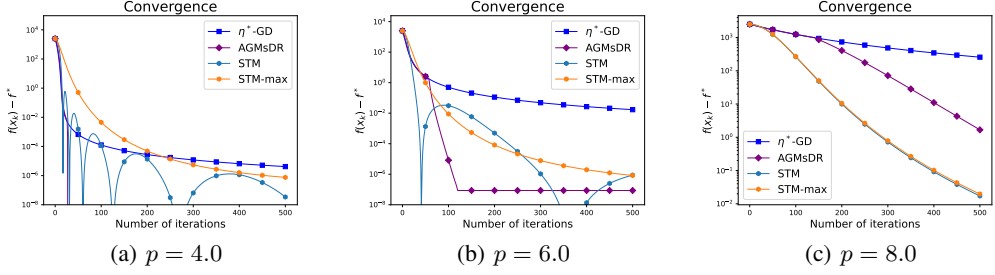

(a) $p = 4.0$        (b) $p = 6.0$        (c) $p = 8.0$

Figure 3: Comparison of Algorithm 1 denoted by AGMsDR with Similar Triangles Method (SMT) and Similar Triangles Method Max (STM-max) for $f(x) = \frac{1}{p}\|x\|^p$, with different values $p$.

