# OpenReview forum: "Optimizing $(L_0, L_1)$-Smooth Functions by Gradient Methods"
_ICLR.cc/2025/Conference — ICLR 2025 Poster_

### Official Review · Reviewer_R7w6 · 2024-10-29

**Soundness:** 3
**Presentation:** 3
**Contribution:** 3
**Rating:** 6
**Confidence:** 5

**Summary:**

The paper examines the extension of a certain optimization method to a broader smoothness assumption, specifically $L_0,L_1$-smoothness. While it broadens the class of functions to be minimized, the authors manage to recover convergence rates comparable to those of the standard $L$-smooth case, albeit with an additional term that does not depend on the final accuracy.

**Strengths:**

- For convex problems, authors improves existing results by achieving $O( L_0R^2 / \varepsilon + L^2_1 R^2)$
- Normalized gradient method and gradient method with Polyak stepsizes, which do not require the knowledge of the parameters $L_0$, $L_1$ and have the same rate as above
- Accelerated method with $O( \sqrt{L_0R^2 / \varepsilon} + L^2_1 R^2)$

**Weaknesses:**

- Lack of analysis  for stochastic case is significantly limiting practical value of the work
- Accelerated method requires a line search (minor)
- The second part of the second question from below

**Questions:**

- As a contribution authors listed in line 54 "tighter bounds on descent inequalities". But in line 58 authors "achieve the best-known" rate for non convex functions. I am curious if the tighter bounds actually helps to improve convergence. I would expect improving the rate, if they actually help. As far as I understand, the tighter bounds are supposed to help for the non-simplified stepsize version and some logarithmic improving factor should appear. Am I right? Could the authors answer this question by providing rates with their tighter bounds and with non-tight bounds?
- Acceleration does not seems fair. By initially running non-accelerated method authors enter the area where $\|\|\nabla f(x) \|\|\leq \frac{L_0}{L_1}$, making the right hand side of the generalized smoothness condition (line 121) $L_0 + L_1|\|\nabla f(x) \|\| \leq 2L_0$, that corresponds to the standard smoothness assumption.
   - In the standard $L$-smooth case there exist accelerated methods without line search (e.g. Similar Triangles Method), that keep iteration bounded ($\|\|x^k - x^* \|\|  \leq R \cdot const$), so I am expecting such methods stay in the area. Can the line search be avoided in your analysis? Or what is it crucial role?

  - The term  $L^2_1 R^2$ corresponds to the number of iteration to enter the area with bounded gradient. Then the authors in fact deal with the standard $2L$-smooth case, while obtaining the other term. This observation is consistent with non-accelerated methods, considered in the paper. I would like to note, that acceleration in fact is used only in the area where the standard $2L$-smooth case takes place, and does not help to enter the area faster. In other words I would say that the contribution in fact is -- a bound on the number of oracle calls to enter this area. And I am expecting this bound to be improved in the accelerated case. Don't get me wrong, but I could not consider the convergence of AGMSDR in the standard $2L$-smooth case as a contribution. I would like authors to address my concerns, and clarify about their contribution to this part. Is it an application of the authors result of non-accelerated method to bound the number to enter the area with bounded gradient?

I would like to comment on the scores. Taking the above questions into account, I currently accept the contributions related to first-order methods. The results are new and significantly improve previous rates. But they rely on the only basic result -- descent inequalities. The rest is an application of the new inequalities to the well-known methods. That is why I consider the contribution as limited. I open for a discussion and the authors are likely to expect higher scores if my concerns are convincingly addressed, especially about the area with bounded gradient.

---

> ### Author Response · Authors · 2024-11-25
>
> Thank you for your feedback.
>
>
> We want to start by highlighting some challenges in developing an accelerated method for $(L_0, L_1)$-smooth functions.
>
> - Firstly, it is not enough to find a good starting point $x_0$ such as $||\nabla f(x)|| \leq \frac{L_0}{L_1}$, because previously considered STM, NAG, and all other existing versions of the standard FGM do not guarantee a monotonic decrease of the gradient norms. As a result, the gradient norm can increase during optimization, and these methods can escape good regions with bounded Hessian norms.
>
> - Secondly, it is not enough to keep iterates inside $||x - x^*|| \leq R \cdot const$ because the iterates might not stay in the initial sublevel set. This also might lead to an increase of gradient norms during the optimization and consequently might lead methods, such as STM, to leave the region where $||\nabla^2 f(x)|| \leq 2 L_0$. The need to upper bound the gradient consequently leads to an additional $\exp(L_1 R)$ (by Lemma 2.2 in our paper) factor in the complexity.
>
> - Instead, we propose to keep iterates inside a good sublevel set by running a method with monotonically decreasing function values. While the idea of staying in a sublevel set is ``simple,'' the complexity of the proposed procedure is **significantly** better than the existing results. Moreover, this idea might be intuitively simple, but the implementation of such an idea turns out to be a non-trivial task for FGM; however, it is possible with careful analysis. To implement it,  we have to (1) identify a good region where iterates would stay, (2) modify the original proof of AGMsDR to work with any update rule $T(\cdot)$, (3) find a general condition for operation $T(\cdot)$ to make proof of AGMsDR work.
>
> - Finally, we did not explain the idea clearly in the initial submission, but now we rewrote the explanations and clearly explained the problem of maintaining the gradient norm small. For details, please see Section 6 and Appendix F.
>
> Next, we provide answers to the reviewer's questions.
>
> 1. Answer to the second question. We kindly disagree that the accelerated procedure is unfair. Even when NAG from [7] and STM from [3] start with a point $x_0$ such that $f(x_0) - f^* \leq \frac{L_0}{5 L_1^2}$, STM requires  $\exp(O(1) L_1 R) \sqrt{\frac{L_0 R^2}{\epsilon}}$  and NAG requires $\mathcal{O}((L_1^2 R^2 +1)\sqrt{\frac{L_0 R^2}{\epsilon} })$ while our procedure (only the second stage, AGMsDR) will require $\mathcal{O}(\sqrt{\frac{L_0 R^2}{\epsilon}})$. It happens since even when starting point $||\nabla f(x_0)|| \leq \frac{L_0}{L_1}$ neither STM nor NAG does not ensure that the gradient norms don't grow, while AGMsDR does. Please see Section 6 and Appendix F for a detailed comparison. Thus, our procedure already achieves much better results than other available methods. That being said, we do agree that our procedure does not accelerate the $O(L_1^2 R^2)$ complexity of the first stage, and it is an interesting open question how to improve on this.
>
> 2. Answer to the first question. Yes, the tighter bound helps us to obtain non-simplified stepsizes $\eta_k^*$ and provides an intuition for all three stepsizes: non-simplified, simplified, and clipping. We are not aware of whether it is possible to improve the complexity of the method in the non-convex case.
>
> We believe that we have addressed all the reviewer's concerns and weaknesses in our paper, and we kindly ask the reviewer to reevaluate their rating in light of these revisions.

---

> ### Comment · Reviewer_R7w6 · 2024-11-28
>
> 1. Regarding AGMsDR:
>
> - It is already possible to use any update rule providing the sufficient decrease. I can not see authors obtaining something new here.
> - I checked the proof, it is identical, authors do not even use $L_0, L_1$, it is just $L$ like originally. When the methods enters the $2L$ smooth area, there is nothing hew here. I agree that it is a good observation that gradient norm stays bounded. But I still believe it might be possible to make iterations of STM bounded as well by increasing the number of iterations of the first stage.
> - It would be great to have the only stage method. Is it possible to introduce another $T(\cdot)$ such that the first stage is avoided? Even by the cost of a worse rate?
> - What if one uses the GD update as $T(\cdot)$? What would be the bound?
>
> 2. I feel that the second question was not answered. As far as I understand (Chen et al., 2023) proved a condition which is equivalent to bounds from Lemma 2.2.
> Could the authors answer this question by providing rates with their tighter bounds and with non-tight bounds? Please show what improvements I can get in terms of final bounds.

---

> > ### Author Response · Authors · 2024-11-30
> >
> > Dear Reviewer,
> >
> > We have already mentioned that the procedure does **significantly** improve the existing results, and you agreed that our procedure is the first to have the gradient norms stay bounded, unlike other methods. We believe this is already an essential contribution to accelerated methods for $(L_0, L_1)$-smooth functions.
> >
> > In general, you seem to critisize our approach for its **simplicity**. However, in our opinion, this cannot be a valid criticism&mdash;being "simple" does not reduce the value of the results if they did not exist before, and especially when several other papers [1,3, 4, 5] couldn't achieve comparable results by using more complicated techniques. On the contrary, we believe that simplicity is a **significant strength** of our approach highlighting its **elegance**.
> >
> > Regarding your specific questions, please see our comments below.
> >
> > > But I still believe it might be possible to make iterations of STM bounded as well by increasing the number of iterations of the first stage.
> >
> > As we mentioned earlier, it is not enough to have bounded iterates because we cannot guarantee that the function value decreases monotonically, making it impossible to bound the gradient norm by a good constant. Indeed, as we already mentioned before, when the distance between the iterates and the optimal solution is bounded by $R$, the best bound we can infer about the norm of the gradient would be of the order of $\exp(L_1 R)$. By increasing the length of the first stage, we cannot unfortunately decrease $R$ compared to its initial value $\| x_0 - x^* \|$ (our function is convex, not strongly convex).
> >
> > > It would be great to have the only stage method. Is it possible to introduce another $T(\cdot)$ such that the first stage is avoided? Even by the cost of a worse rate?
> >
> > Yes, of course. This is exactly what other existing accelerated methods for $(L_0, L_1)$-smooth functions [1, 2] are doing. However, their complexity guarantees are much worse.
> >
> > That being said, we agree that developing a one-stage method with the same complexity bounds as ours is an interesting and important question, which should be addressed in a future work.
> >
> > > What if one uses the GD update as $T(\cdot)$? What would be the bound?
> >
> > Could you please clarify what exactly you mean by the "GD update"? Note that we already allow using the gradient update with any of the stepsizes we introduced in Section 3.
> >
> > >  ... As far as I understand [6] proved a condition which is equivalent to bounds from Lemma 2.2. Could the authors answer this question by providing rates with their tighter bounds and with non-tight bounds? ....
> >
> > The specific inequalities we establish in Lemma 2.2 are tighter than those from Proposition 3.2 in [6] because our error functions $e^t - 1$ and $\phi(t) \equiv e^t - t - 1$ are smaller than $t e^t$ and $\frac{t^2}{2} e^t$, respectively, from [6]. That being said, one can still use the weaker bounds from [6] to achieve the same complexity (up to an absolute constant factor) for most the methods we consider in our paper.
> >
> > An important advantage of our tighter bound compared to that from [6] is that it allows one to obtain an explicit formula for the optimal step size $\eta_*$ by minimizing the tighter upper bound on the function. Indeed, if you try to repeat our derivation of the **optimal step size** (at the beginning of Section 3), you will see that, when using the bound from [6], you need to minimize $\gamma t - \frac{t^2}{2} e^t$ (in $t$) which cannot be solved in closed-form (in contrast to ours $\gamma t - [e^t - t - 1]$). Introducing the optimal stepsize and the corresponding bound on the function progress, besides being interesting in itself, allows us to clearly explain why the **simplified** and **clipping step sizes** are indeed natural for $(L_0, L_1)$-smooth functions (because they approximate the optimal step size by preserving the same bound on the function progress, up to an absolute constant factor).
> >
> > **References:**
> >
> > [1] Jingzhao Zhang, Tianxing He, Suvrit Sra, and Ali Jadbabaie. Why gradient clipping accelerates training: A theoretical justification for adaptivity. ICLR 2020.
> >
> > [2] Eduard Gorbunov, Nazarii Tupitsa, Sayantan Choudhury, Alen Aliev, Peter Richtarik, Samuel Horvath, and Martin Takac. Methods for convex $(L_0, L_1)$-smooth optimization: Clipping, acceleration, and adaptivity. ArXiv, 2024.
> >
> > [3] Anastasia Koloskova, Hadrien Hendrikx, and Sebastian U Stich. Revisiting gradient clipping: Stochastic bias and tight convergence guarantees. ICML 2023.
> >
> > [4] Haochuan Li, Jian Qian, Yi Tian, Alexander Rakhlin, and Ali Jadbabaie. Convex and non-convex optimization under generalized smoothness, NeurIP, 2023.
> >
> > [5] Yuki Takezawa, Han Bao, Ryoma Sato, Kenta Niwa, and Makoto Yamada. Polyak meets parameter free clipped gradient descent, NeurIPS, 2024.
> >
> > [6] Ziyi Chen, Yi Zhou, Yingbin Liang, and Zhaosong Lu. Generalized-smooth nonconvex optimization is as efficient as smooth nonconvex optimization, ICML, 2023.

---

> > > ### Comment · Reviewer_R7w6 · 2024-11-30
> > >
> > > >Could you please clarify what exactly you mean by the "GD update"? Note that we already allow using the gradient update with any of the stepsizes we introduced in Section 3.
> > >
> > > I mean what would the bound if you run AGMsDR without the first stage, just starting from $x^0$ with the gradient step as $T(\cdot)$ (which is the step for the first stage GD)? AGMsDR is still monotone, like GD. As far as I understand AGMsDR works not worse that GD for the standard smoothness assumption, can the same be said about generalized smoothness? If it can it seems to me that the first stage can be avoided.
> > >
> > > The reason I think it is important is that your two stage procedure need knowing constants $L_0, L_1$ to determine the number of steps for the first stage (or maybe at least their ration for sufficiently and an upper bound on $L_0$ and $L_1$). The latter means that the methods needs two parameters in contrast to GD, which heeds only one: $L$ such that $L \geq L_0$ and $L \geq L_0$.
> > >
> > > Having one stage method significantly increases  the importance of your result for acceleration.
> > >
> > > >achieve the same complexity (up to an absolute constant factor)
> > >
> > > Now I see you point regarding tighter bounds, please consider clarifying this contribution in the final version. But I am still interested in constants. Can you estimate the constant?

---

> > > > ### Author Response · Authors · 2024-12-01
> > > >
> > > > > I mean what would the bound if you run AGMsDR without the first stage, just starting from x0 with the gradient step as $T(\cdot)$ (which is the step for the first stage GD)?
> > > >
> > > > Ok, we see now what you mean. The corresponding bound is $\mathcal{O}(\sqrt{\frac{(L_0 + L_1^2 F_0) R^2 }{\epsilon}})$, and is valid for any starting point $x_0$ (with $F_0 \equiv f(x_0) - f^*$). This estimate of one-stage AGMsDR is already much better than the $\mathcal{O}([L_1^2 R^2 + \frac{L_1^2 F_0}{L_0} + 1] \sqrt{\frac{L_0 R^2 + F_0}{\epsilon}})$ complexity of NAG from [4]. When we start AGMsDR from a "good" starting point $x_0$ satisfying $F_0 \leq \mathcal{O}(\frac{L_0}{L_1^2})$, the complexity of AGMsDR becomes $\mathcal{O}(\sqrt{\frac{L_0 R^2}{\epsilon}})$ which is exactly what we already wrote in the paper. But we agree with you that it would indeed be better to first present the general result for AGMsDR which is valid for any starting point, and only then propose our two-stage procedure as a simple improvement. We will adjust the presentation accordingly in the final version of the paper.
> > > >
> > > > > As far as I understand AGMsDR works not worse that GD for the standard smoothness assumption, can the same be said about generalized smoothness?
> > > >
> > > > This is a bit tricky. At the moment, we can only prove the following complexities for GD and AGMsDR, respectively: $\mathcal{O}(\frac{L_0 R^2}{\epsilon} + L_1 R \ln \frac{F_0}{\epsilon})$ and $\mathcal{O}(\sqrt{\frac{(L_0 + L_1^2 F_0) R^2 }{\epsilon}})$. Therefore, without doing anything else, we cannot say, unfortunately, that AGMsDR is always better than GD. Understanding how to fix this drawback is definitely an interesting question for future research.
> > > >
> > > > >  But I am still interested in constants. Can you estimate the constant?
> > > >
> > > > It is difficult to estimate it exactly but it will be around $4$ or $6$ for GD with our simplified or clipped stepsizes.
> > > >
> > > > **References**
> > > >
> > > > [4] H. Li, J. Qian, Y. Tian, A. Rakhlin, and A. Jadbabaie. Convex and nonconvex optimization under generalized smoothness.

---

> > > > > ### Comment · Reviewer_R7w6 · 2024-12-02
> > > > >
> > > > > Thanks, I still think that the accelerated result is less practical than the non-accelerated, because good initialization is required. But I am updating my score.

---

> > > > > > ### Author Response · Authors · 2024-12-04
> > > > > >
> > > > > > Dear reviewer,
> > > > > >
> > > > > > We thank you for your time, helpful feedback, and fruitful discussion, which led to the improvement of our paper. We are also delighted by the positive evaluation of our paper and increased score.

---

### Official Review · Reviewer_KuU4 · 2024-10-31

**Soundness:** 3
**Presentation:** 3
**Contribution:** 3
**Rating:** 6
**Confidence:** 3

**Summary:**

The author(s) present non-asymptotic convergent analyses of various types of gradient methods on convex and non-convex functions satisfying $(L_0, L_1)$-smoothness, which is a type of generalized smoothness assumption that has gained some interests recently in machine learning community.

They derive a series of first-order upper bounds, implied by the generalized smoothness assumption, which are tigher bounds than previous works. Furthermore, they derive similar lower bounds when the functions are convex.

Using these newly derived bounds, they present a convergent analysis on non-convex functions, which matches the current best-known rate of this function class up to absolute constants.

For convex functions, however, they derive better-than-existing convergent rates for different gradient methods of this particular function class.

Finally, they experiment with a simple $||\mathbf{x}||^p / p$ function, where the constants $L_0$ and $L_1$ can be known in advance.

**Strengths:**

To me, one of this paper's main strengths/contributions is the derivation of tighter first-order upper and lower bounds. The bounds will undoubtedly help any further work on analyzing this particular function class.

In Theorems 3.2 and 5.1, the author(s) is able to dispense $L$-gradient Lipschitz, which was additionally assumed in the previous works Koloskova et al 2023, while improving the current existing rate.

The overall flow of the paper is well-presented.

**Weaknesses:**

1. Numerical results can be strengthened by providing more experiments.

2. For Theorem 3.1, even though your rate is better than the rate provided in Hubler et al 2024, they do not have the dependency on $L_0$ and $L_1$ (See section Questions, 1.).

3. (line 269) I believe "By Theorem 3.1, the smallest number K of iterations required to achieve ..." is incorrect, as one can happen to choose an initial point such that $\|\| \nabla f(x_{0}) || < \epsilon_{\mathbf{g}}$. In this case, $K = 0$, which violates your claim here. The correct wording should be "By Theorem 3.1, obtaining the stationary point requires at most ... ".

4. (Theorem 5.1) The presentation of your theorems should be consistent. It is recommended to present in the form of "$K \geq $", like in your other theorems.

5. $||\mathbf{x}_0 - \mathbf{x}^*||$ is denoted to be both $R_0$ and $R$, e.g., Theorem 3.2. Notations should be consistent.

6. (line 257) "absolte" -> "absolute"

7. (eq. 14) I think the inequality should be $... \geq \frac{||\nabla f(x)||^2}{2L_0 + 3L_1||\nabla f(x)||}$, i.e., there is no 2 in the numerator (c.f. Lemma B.1)

8. (line 641) I think $s$ should be $s := \nabla f(x) - \nabla f(y)$. Otherwise, the sign in the inequality at line 643 is incorrect.

9. (line 648) Change $\phi^*(\gamma)$ to $\phi_*(\gamma)$ to maintain consistency.

10. (line 651) Some of the gradients should be of variable $y$, not just $x$, i.e., $\nabla f(y)$.

**Questions:**

1. Another way to dispense of the dependency of the constants $L_0$ and $L_1$ in the step-size is to perform backtracking line-search, which also guarantees descents in the objective function value. Can the given new bounds (Lemma 2.2, 2.3) improve the rate derived by Hubler et al 2024?

2. For Theorem 6.1, empirically, is there any way to tell when to switch to AGMsDR? That is, can we tell that we are in a local region?

3. Have you tried experimenting with a softmax function, which I believe the constants $L_0$ and $L_1$ can be analytically evaluated? This example will provide more insights than the simple function $||\mathbf{x}||^p$.

---

> ### Author Response · Authors · 2024-11-25
>
> Thank you for the positive evaluation of our paper and feedback. We incorporated all the writing mistakes and typos in the updated version of our paper. Please see Global Response and the updated version. Below we provide answers to reviewer's concerns
>
> - Regarding comparison with Hübler et al. (2024). The rate in Hübler et al. 2024 depends on $L_0, L_1$, but to run the gradient method with backtracking line search, they don't need to know $L_0, L_1$. We added this comment in the updated version of our paper.
>
> - Regarding line 641. It does not matter if $s := \nabla f(x) - \nabla f(y)$ or $s := \nabla f(y) - \nabla f(x)$, since the sign is included inside of $h \in \mathbb{R}^m$ which we maximize over.
>
> Next, we provide answers to the reviewer's questions.
>
> 1. Answer to the first question. We don't think that the complexity of the gradient method with backtracking line search, as presented in Hübler et al. 2024, can be improved using our tighter bound.
>
> 2. Answer to the second question. Our accelerated procedure requires some extra parameters to switch to the second stage. We assume that we can check that $f(x) - f^* \leq \Delta$ for any given $\Delta$. Another option is to run GM for $K=O(L_1^2 R^2)$ number of iterations if we know $L_1$ and $R$); then the inequality $f(x) -f^*  \leq \Delta$ is guaranteed to be satisfied by our Theorem 3.2. Developing an adaptive procedure with the same complexity is an interesting open question for future research.
>
> 3. Answer to the third question. We are not aware of any results explicitly showing parameters $L_0, L_1$ for the softmax function. Unfortunately we were not able to compute constants $L_0, L_1$ explicitly for the softmax function.
>
> Regarding experiments. The focus of this paper is mostly theoretical and numerical experiments are conducted to verify our theoretical findings. There are already many works with experiments showing an advantage of clipped Zhang et al. (2019), normalized methods Hübler et al. (2024), and Polyak-stepsizes Takezawa et al. (2024). However, in the updated version of our paper, we provided more experiments comparing accelerated methods to support our theoretical findings.
>
>
> We believe that we have addressed all the reviewer's concerns and weaknesses in our paper, and we kindly ask the reviewer to reevaluate their rating in light of these revisions.
>
> [Zhang et al. (2019)] Jingzhao Zhang, Tianxing He, Suvrit Sra, and Ali Jadbabaie. Why gradient clipping accelerates
> training: A theoretical justification for adaptivity. ICLR 2020.
>
> [Hübler et al 2024] Florian Hubler, Junchi Yang, Xiang Li, and Niao He. Parameter-agnostic optimization under relaxed ¨
> smoothness. AISTATS, 2024.
>
> [Takezawa et al 2024] Yuki Takezawa, Han Bao, Ryoma Sato, Kenta Niwa, and Makoto Yamada. Polyak meets parameter free clipped gradient descent, NeurIPS, 2024.

---

> > ### Comment · Reviewer_KuU4 · 2024-12-02
> >
> > I appreciate for the author(s)' reply. I am satisfied with their comments and corrections to the paper. I think it is a decent paper and should be accepted. However, I will still stand by my previous evaluations.

---

> > > ### Author Response · Authors · 2024-12-04
> > >
> > > Dear reviewer,
> > >
> > > We thank you for your time, helpful feedback, and fruitful discussion, which led to the improvement of our paper. We are also delighted by the positive evaluation of our paper.

---

### Official Review · Reviewer_gEoz · 2024-10-31

**Soundness:** 3
**Presentation:** 3
**Contribution:** 2
**Rating:** 6
**Confidence:** 4

**Summary:**

The paper studies gradient methods for solving problems with (L0, L1)-smoothness, a recent class of functions developed by Zhang et al. (2019). Novel convergence results are established in this paper, including tighter bounds on function growth and the complexity bound for finding near stationary points in both nonconvex and convex cases. Some new methods with different step sizes are also examined.

**Strengths:**

1. The paper is well-written and easy to follow. The analysis is thorough and offers many insights for research in optimization.

2. I appreciate the comparison between this paper and Gorbunov et al. (2024), which is comprehensive and particularly crucial for readers.

3. I have attempted to implement some of the suggested algorithms and found that the analysis presented here is essential for practical use, especially when the optimal step size in equation (11) is given explicitly.

**Weaknesses:**

The following are some major concerns about the paper.

1. The numerical experiments are conducted only on simple functions. I suggest that the authors significantly improve this section by employing more diverse examples, particularly in nonconvex cases. For examples, could the authors consider loss functions from deep learning models or other challenging nonconvex optimization problems from applications?

2. If I have not overlooked anything, the current paper does not discuss the difficulties of deriving the convergence properties for (L0, L1)-smooth functions compared to L-smooth functions, nor how the authors overcome these challenges. Can the author discuss key challenges in analyzing (L0,L1)-smooth functions compared to L-smooth functions, and to highlight any novel theoretical tools or techniques they developed to overcome these challenges.

3. I do not find Proposition 2.6 impressive, as the results are rather predictable, and the proofs seem straighforward.

4. The authors mention that "we cannot guarantee that the class is closed under all standard operations, such as the summation". Can you provide a concrete counterexample showing that the sum of two (L0,L1)-smooth functions is not necessarily (L0,L1)-smooth? If so, are there any additional conditions that could guarantee the sum remains in this function class?

5. The authors claim that the estimate in (4) is tighter than those presented in previous works. Can you provide a numerical example or theoretical argument demonstrating how your estimate in (4) is tighter than previous bounds? Is it possible to prove that this bound is optimal, or are there cases where it could potentially be improved further?

**Questions:**

See weakness

---

> ### Author Response · Authors · 2024-11-25
>
> Thank you for your feedback and recommendations. Below we provide answers to reviewer's concerns
>
>
> 1. Regarding numerical experiments. We want to highlight that most of our results are in a convex setting where we significantly improved the best-known complexity. The main focus of this paper is on theoretical insights. There are already many works with experiments on non-convex setting showing an advantage of clipped [Zhang et al. (2019)], normalized methods [Hübler et al. (2024)] and polyak-stepsizes [Takezawa et al. (2024)]. To address this concern, we provided more experiments comparing accelerated methods to support our theoretical findings.
>
> 2. Regarding challenges in analysis of $(L_0, L_1)$-smooth functions. Our work is not the first paper focusing on $(L_0, L_1)$-smooth functions. The previous works discussed the difficulty of analyzing $(L_0, L_1)$-smooth functions (see Zhang et al 2019, Koloskova et al 2023, Li et al 2023, Hübler et al 2024). We don't provide such a discussion since it was already done previously and due to space limitations.
>
> 3. Regarding Proposition 2.6. In our paper, we are not claiming that Prop 2.6 is impressive. However, these results might be important for a community to investigate further properties of $(L_0, L_1)$-smooth functions. We believe that it's useful to have such results explicitly written somewhere to streamline their application. We also wanted to attract the attention of the readers to the problem of insufficient calculus rules for this class of functions (problems with summation and affine substitution of the argument).
>
> 4. Regarding closedness under summation. We do agree that finding such an example is interesting and important. But we are not claiming that it is not closed; in our paper, we wrote that we could not guarantee that. Please note that Proposition 2.6 (Proposition A.3 in the updated version) provides a condition when the sum of two $(L_0, L_1)$-smooth function is preserved: $f$ is $(L_0, L_1)$-smooth, $g$ is $(L, 0)$-smooth and is Lipschitz continuous, and their sum $f+g$ is $(L_0 + M L_1 +L , L_1)$-smooth. However, even under this condition the resulting parameter $L_0 + M L_1 +L$ is larger than $\max \{L_0, L\}$.
>
> 5. Regarding tighter bounds. Our estimate in the first inequality of Lemma 2.5 is tighter than inequality (13) in Corollary A.4 [Zhang et al 2020], since ours $(e^t - 1)/t \leq 1 + e^t - (e^t - 1)/t$ for $t \geq 0$. Our second bound in Lemma 2.5 is tighter than the bound in eq (7) in Lemma A.3 in Zhang et al 2020 since our $\phi(t) = e^t - t -1 \leq (e^t - 1) t \leq 1 + e^t - (e^t - 1)/t$.
>
> We believe that we have addressed all the reviewer's concerns and weaknesses in our paper, and we kindly ask the reviewer to reevaluate their rating in light of these revisions.
>
> [Zhang et al. (2019)] Jingzhao Zhang, Tianxing He, Suvrit Sra, and Ali Jadbabaie. Why gradient clipping accelerates
> training: A theoretical justification for adaptivity. ICLR 2020.
>
> [Zhang et al 2020] Bohang Zhang, Jikai Jin, Cong Fang, and Liwei Wang. Improved analysis of clipping algorithms for
> non-convex optimization, NeurIPS, 2020.
>
> [Koloskova et al 2023] Anastasia Koloskova, Hadrien Hendrikx, and Sebastian U Stich. Revisiting gradient clipping:
> Stochastic bias and tight convergence guarantees, ICML, 2023
>
> [Li et al 2023] Haochuan Li, Jian Qian, Yi Tian, Alexander Rakhlin, and Ali Jadbabaie. Convex and non-convex
> optimization under generalized smoothness, NeurIP, 2023.
>
> [Hübler et al 2024] Florian Hubler, Junchi Yang, Xiang Li, and Niao He. Parameter-agnostic optimization under relaxed ¨
> smoothness. AISTATS, 2024.
>
> [Takezawa et al 2024] Yuki Takezawa, Han Bao, Ryoma Sato, Kenta Niwa, and Makoto Yamada. Polyak meets parameter free clipped gradient descent, NeurIPS, 2024.

---

> > ### Comment · Reviewer_gEoz · 2024-12-01
> >
> > Thanks for your clarifications and explanations. I am increasing my score, leaning a bit towards acceptance for the paper.

---

> > > ### Author Response · Authors · 2024-12-04
> > >
> > > Dear reviewer,
> > >
> > > We thank you for your time, helpful feedback, and fruitful discussion, which led to the improvement of our paper. We are also delighted by the positive evaluation of our paper and increased score.

---

### Official Review · Reviewer_qpgv · 2024-11-02

**Soundness:** 4
**Presentation:** 2
**Contribution:** 3
**Rating:** 8
**Confidence:** 3

**Summary:**

This paper provides a general understanding of the $(L_0, L_1)$-smooth function class, which has recently gained increasing interest due to its empirically observed connections to deep learning applications. They authors establish a parallelism between the classical optimization theory for $L_0$-smooth functions, and show how analogous inequalities and algorithm design strategies should be carried out with $(L_0, L_1)$-smooth functions. This includes the derivation of $(L_0, L_1)$-smooth version of the cocoercivity inequality, analysis of fixed-stepsize gradient descent method for convex and nonconvex cases, and analysis of gradient methods using normalized stepsizes, Polyak stepsizes, and Nesterov-type acceleration for convex functions.

**Strengths:**

As mentioned in the Summary section, this paper nicely demonstrates the parallelism of how the traditional techniques from smooth (convex) minimization should be properly interpreted and applied to the class of $(L_0, L_1)$-smooth functions. As a reader who is much more familiar with the classical optimization theory but not as much with the recent theory of $(L_0, L_1)$-smooth optimization, the paper was interesting, easily readable, and informative. It seems such viewpoint has not been provided previously with this level of clarity, and I think this work could serve as a solid reference for those future readers who are interested in this area. The algorithms and their convergence results the paper provides within various setups seem correct and competitive.

**Weaknesses:**

The primary weakness of the current version of the paper is the writing. While I think the technical contents are good, the writing does not seem to be polished carefully enough and should be managed before the publication. The general flow is okay, but there are many detailed points which I would recommend the authors to address; please refer to the Questions section. I would recommend acceptance of the paper provided that the writing issues get resolved.

**Questions:**

Writing issues (in the order of appearance)
- In pg. 3, lines 108-109, it will be illustrative to explain in more detail in which aspects the authors' proof techniques differ from that of the work [1], cited as Gorbunov et al. (2024) in the manuscript. Also I think the authors should be more cautious about claiming that the proof is more "elegant" than something else. One possible suggestion is to emphasize their clearer parallelism with the classical theory, rather than to describe the value of this paper's framework with a subjective expression like "elegant".
- The statement of Lemma 2.2 involves the notation $\phi(t) = e^t - t - 1$, while its proof in the Appendix uses the symbol $\phi$ to denote something else. Please change it.
- The bound $\phi_* (\gamma) \ge \frac{\gamma^2}{2+\gamma}$ appears suddenly in the statement of Lemma 2.3, while the relevant discussion appears later in Section 3. I think this should be relocated to somewhere near Lemma 2.3.
- In pg. 4, lines 164-165, the meaning of the sentence "we present some examples to support the choice of generalized smooth assumption" is unclear.
- Proposition 2.6 and the subsequent discussion doesn't seem very relevant to the rest of the paper. To some readers this information could be valuable, but generally speaking, I even think the whole part can be moved to the Appendix without altering the flow of the paper. Also, please clearly define the expressions like "$M$-Lipschitz twice continuously differentiable" before using it.
- In line 183-184, there is dangling "for some" after the period.
- In Theorem 3.1, it should have been assumed/defined that $f^* := \inf f > - \infty$. In general, the authors did not properly state that they assume the existence of a minimizer $x^*$, etc. (in subsequent theorems regarding convex cases). Please be precise about these points.
- In the statement of Theorem 3.2, is $||x_0 - x^*||$ intended to be denoted as $R_0$?
- In pg. 6, in lines 297-298, it is not clear what the authors mean with "by using the preceding estimate and the update rule of the method". Only after reading the proof in the Appendix I could understand the technique deriving equation (20), and I still do not feel that the text surrounding the equation is successfully conveying the idea. If you intend to provide a high-level intuition, please be more precise and detailed. Also, note that the exponent 2 is missing in $R_{k+1}^2$ within (20), and in equation (43), the $g_k^2$ factors seem to be missing.
- Lines 304-310 provide an interesting insight on the complexity $\mathcal{O}(L_0 R_0^2 / \epsilon + L_1^2 R_0^2)$, but I think the complexity analysis does not clearly show that the constant $\mathcal{O}(L_1^2 R_0^2)$ complexity corresponds to the phase where the algorithm reaches the regime where $f$ behaves like a smooth function. (Which is in contrast with Section 6, where the algorithm is deliberately designed in this way.) Can this discussion somehow be made more precise?
- In Theorem 4.2, $R$ is used without being defined.
- I do not see any reason why Lemma 4.1 is introduced in pg. 7. I think a discussion on how (22) is utilized in the analysis is missing.
- In lines 349-350, the authors say "time-varying stepsizes can be eliminated", but this seems to be a conjecture (or at least, something that the authors do not formally prove) and I think it should be reworded. Please be careful about making such statement.
- In pg. 8, lines 378-380, the authors emphasize that their rate does not have an exponential dependency on $L_0, L_1$, but why does this worth emphasis? Are there prior work which analyzed gradient method with Polyak stepsizes in the same setup and obtained rate that involves $\exp (L_0), \exp (L_1)$? If so, please specify.
- In lines 400-401, the authors mention that "Our rate is better than $\mathcal{O}(L_0 R^2 / \epsilon + \sqrt{L / \epsilon} L_1 R^2)$", but where does this rate come from?
- In the beginning of Section 6 (and also the same for Sections 4, 5) please clearly specify in the first sentence introducing the setup that you are dealing with convex functions.
- It is weird that Algorithm 1 (which is not the algorithm that the authors develop) is packaged using the Algorithm environment, while the authors' algorithm (lines 427-431) isn't.
- The Step 1 of the authors' algorithm in Section 6 requires to find a point $\bar{x}$ satisfying $f(\bar{x}) - f(x^*) \le \frac{L_0}{8 L_1^2}$. I think this requires either the knowledge of $f(x^*)$ or $R=||x_0 - x^*||$ (in order to determine when to stop). Is this understanding correct? If so, this is a limitation that should be clearly explained.
- Theorem 6.1 misses the convexity assumption in the statement. Also, it analyzes the AGMsDR algorithm with initial point satisfying  $f(x_0) - f(x^*) \le \frac{L_0}{8 L_1^2}$, but this is inconsistent with the $\bar{x}$ notation used in lines 427-431. I think this also comes from the problem of not highlighting their "procedure" as a separate algorithm with a proper name.
- In lines 477-478, should the two occurrences of $R$ be the same values? It seems like one is $||x_0 - x^*||$ and one is $||\bar{x} - x^*||$.
- In line 483, the authors again mention "not have an exponential dependency". Please cite a relevant prior work explaining why this is worth emphasis.
- It is not clearly defined what the figure labels like NGD, GD1, GD2, etc. precisely means. Even though it could be speculated if one reads through Section 7, this should be done somewhere, for example, within the figure caption.
- In Figure 2, please specify that you are using $L_0 = (\frac{p-2}{L_1})^{p-2}$ according to Example 2.4 (if this understanding is correct).

[1] Gorbunov et al., Methods for Convex $(L_0,L_1)$-Smooth Optimization: Clipping, Acceleration, and Adaptivity.

---

> ### Author Response · Authors · 2024-11-25
>
> Thank you for the positive evaluation of our paper and a detailed list of recommendations regarding our paper's writing. We significantly improved our new version, given all the reviewer's recommendations. Please see our global response and updated version of our paper for details.
>
> Regarding the knowledge of $f^*$ for the accelerated two-stage procedure (Algorithm 6.1). You are right; Algorithm 6.1 requires prior knowledge of $f^*$. We added this discussion. Another option is to run GM for $K = O(L_1^2 R^2)$ iterations; please see our response to reviewer KuU4.
>
> We believe that we have addressed all the reviewer's concerns and weaknesses in our paper, and we kindly ask the reviewer to reevaluate their rating in light of these revisions.

---

> > ### Author Response · Authors · 2024-12-02
> >
> > Dear Reviewer,
> >
> > We believe we have thoroughly addressed all of the reviewer's concerns and suggestions in the updated version of our paper. Given that the deadline for the authors-reviewers discussion is approaching, we kindly request the reviewer to reevaluate their rating in light of these revisions.

---

> > > ### Comment · Reviewer_qpgv · 2024-12-03
> > >
> > > The authors have indeed addressed most of the presentation-related issues with a number of major changes to the manuscript, and I appreciate the effort. Although I currently do not see any objection from other reviewers toward accepting this paper, I will raise my score to provide a stronger support.
> > >
> > > I do want to mention that there are some unaddressed comments which I would recommend the authors reflect on in the camera-ready version.
> > > - In the beginning of Section 6, please state that you are developing acceleration for $(L_0, L_1)$ smooth **convex** function.
> > > - While the authors now mention in line 508 that "However, Algorithm 2 requires additional knowledge of $f^*$ to stop the first stage of the procedure.", I do not think this is a proper way of acknowledging limitation. This should be more clearly explained at a much earlier stage, as technically, it is an acceleration *under additional knowledge*, and currently the emphasis is really minimal. Please appropriately expand the discussion on this point.
> > >
> > > Minor: Please check the grammar of lines 505~507 (the sentence starting with "While").

---

> > > > ### Author Response · Authors · 2024-12-04
> > > >
> > > > Dear reviewer,
> > > >
> > > > We thank you for your time, helpful feedback, and fruitful discussion, which led to the improvement of our paper. We are also delighted by the positive evaluation of our paper and increased score. We will address the remaining comments in the final version of our paper.

---

### Author Response · Authors · 2024-11-25
**Global Response**

We thank all the reviewers for their time and suggestions. We are delighted that most of you found our work interesting, insightful, and correct.

Based on the obtained feedback, we have prepared a revised version of the manuscript where we significantly improved the paper's structure and fixed many typos.
Please see the updated PDF file.
Below is a quick summary of the most important changes (highlighted in blue in the revised version):

1. We have considerably improved the writing and explanations. For instance, in Section~3, we now more carefully introduce and compare the three stepsizes. Also, we rewrote the explanations for the accelerated method to highlight the main idea and challenges. Please see Section 6 and Appendix F for more details.

2. We have modified the proof of Theorem 3.2 for the Gradient Method (GM) in the convex case and obtained the $\mathcal{O}(\frac{L_0 R^2}{\epsilon} + L_1 R \ln \frac{F_0}{\epsilon})$ complexity which is better than the previous $\mathcal{O}(\frac{L_0 R^2}{\epsilon} + L_1^2 R^2)$, especially when $F_0$ is reasonably bounded. Please see Section 3 for more details.

3. We have added the formal proof of the fact that the Normalized Gradient Method (NGM) with time-varying stepsizes obtains the same complexity as that with fixed stepsizes, up to an extra logarithmic factor. Please see Appendix C.3 for more details.

4. We have also clarified, in Section 4, that NGM does not need to know the exact value of $R$ and converges with any $\hat{R}$ at the expense of multiplying the complexity by the squared ratio of $\hat{R}$ and $R$. We have also elaborated on the proof idea behind Theorem 4.1.

Finally, we would like to point out a few important aspects of our work that some of the reviewers might have missed.

First, we provide many useful insights for the optimization of $(L_0, L_1)$-smooth functions. In particular, in Section 3, we show, for the first time, that the popular clipping stepsize is indeed natural for $(L_0, L_1)$-smooth functions, and is nothing but an (accurate) approximation of the optimal stepsize resulting from minimizing an upper bound on the objective function.

Second, we establish several **completely novel** convergence rate results which are **significantly** better than those available in the literature:

- Our $\mathcal{O}(\frac{L_0 R^2}{\epsilon} + L_1 R \ln \frac{F_0}{\epsilon})$ complexity of GM is much better than the $\mathcal{O}(\frac{(L_0 + L_1 \| \nabla f(x_0) \|) R^2}{\epsilon})$ from [Li et al 2023].
It is also significantly better than the $\mathcal{O}(\frac{L_0 R^2}{\epsilon} + L_1 R \sqrt{\frac{L}{\epsilon}})$ complexity from [Koloskova et al 2023], and, in contrast to the latter work, we do not need to impose any additional (rather strange) assumption that $f$ is also $L$-smooth.

- Our $\mathcal{O}(\frac{L_0 R^2}{\epsilon} + L_1^2 R^2)$ complexity for GM with Polyak stepsizes is significantly better than the only available result of $\mathcal{O}(\frac{L_0 R^2}{\epsilon} + L_1 R \sqrt{\frac{L}{\epsilon}})$ from [Takezawa et al 2024] (which also assumes, as [Koloskova et al 2023], that $f$ is additionally $L$-smooth).

- The $\mathcal{O}(\frac{L_0 R^2}{\epsilon} + L_1^2 R^2)$ complexity we establish for NGM is entirely new and there are no other results for this method in the literature.

- Our $\mathcal{O}(m \sqrt{\frac{L_0 R^2}{\epsilon}} + L_1^2 R^2)$ complexity for the two-stage acceleration procedure is significantly better than the $O((L_1^2 R^2 + \frac{L_1^2 F_0}{L_0} + 1) \sqrt{\frac{F_0 + L_0 R^2}{\epsilon}})$ complexity for NAG from [Li et al 2023], even when the starting point is such that $F_0 \equiv f(x_0) - f^* = O(\frac{L_0}{L_1^2})$, as required for the second stage of our procedure. Please see Section 6 for more details.


[Koloskova et al 2023] Anastasia Koloskova, Hadrien Hendrikx, and Sebastian U Stich. Revisiting gradient clipping:
Stochastic bias and tight convergence guarantees, ICML, 2023

[Li et al 2023] Haochuan Li, Jian Qian, Yi Tian, Alexander Rakhlin, and Ali Jadbabaie. Convex and non-convex
optimization under generalized smoothness, NeurIP, 2023.

[Takezawa et al 2024] Yuki Takezawa, Han Bao, Ryoma Sato, Kenta Niwa, and Makoto Yamada. Polyak meets parameter free clipped gradient descent, NeurIPS, 2024.

---

### Meta-Review · Area_Chair_GJ9J · 2024-12-17

**Metareview:**

The paper studies gradient methods for optimization problems with \((L_0, L_1)\)-smooth objectives, a generalization of Lipschitz-smooth functions. It offers new insights into this class and introduces a unified analysis framework for related methods. The authors establish state-of-the-art complexity guarantees for the simple gradient method in both convex and non-convex settings. They also analyze two variants--Polyak stepsize and normalized gradient methods--achieving optimal bounds without requiring prior parameter knowledge. Additionally, the paper extends accelerated gradient methods to this function class, demonstrating desirable convergence properties.

Overall the reviewers agree that the paper is interesting, contains several novel ideas, and makes notable contributions to the theory of first-order optimization methods.

**Additional Comments On Reviewer Discussion:**

While most reviews are positive, some suggest improving the presentation and strengthening the numerical experiments, which the authors are encouraged to address in their final revision.

---

### Decision · Program_Chairs · 2025-01-22

Accept (Poster)